# A distinct growth physiology enhances bacterial growth under rapid nutrient fluctuations

Jen Nguyen [1,2], Vicente Fernandez[1], Sammy Pontrelli[3], Uwe Sauer [3], Martin Ackermann[4,5] & Roman Stocker [1✉]

It has long been known that bacteria coordinate their physiology with their nutrient environment, yet our current understanding offers little intuition for how bacteria respond to the second-to-minute scale fluctuations in nutrient concentration characteristic of many microbial habitats. To investigate the effects of rapid nutrient fluctuations on bacterial growth, we couple custom microfluidics with single-cell microscopy to quantify the growth rate of *E. coli* experiencing 30 s to 60 min nutrient fluctuations. Compared to steady environments of equal average concentration, fluctuating environments reduce growth rate by up to 50%. However, measured reductions in growth rate are only 38% of the growth loss predicted from single nutrient shifts. This enhancement derives from the distinct growth response of cells grown in environments that fluctuate rather than shift once. We report an unexpected physiology adapted for growth in nutrient fluctuations and implicate nutrient timescale as a critical environmental parameter beyond nutrient identity and concentration.

[1] Institute of Environmental Engineering, ETH Zürich, Zürich, Switzerland. [2] Microbiology Graduate Program, Massachusetts Institute of Technology, Cambridge, MA, USA. [3] Institute of Molecular Systems Biology, ETH Zürich, Zürich, Switzerland. [4] Institute of Biogeochemistry and Pollutant Dynamics, ETH Zürich, Zürich, Switzerland. [5] Department of Environmental Microbiology, Eawag, Dübendorf, Switzerland. ✉email: romanstocker@ethz.ch

Our planet is sustained by the metabolic activities of microorganisms. In our gut, microbial communities break down nutrients into forms that we can take up and use; at sea, microbial growth affects the sequestration of carbon in the ocean and its release back into the atmosphere; and in the soil, microbes convert organic molecules into forms that facilitate plant growth. These metabolic activities are often performed under conditions that depart from a steady state. Rather, the quality and quantity of available nutrients often fluctuate rapidly due to microscale spatial heterogeneity, fluid flow, or host eating habits. Many host-associated and free-living microbes experience second- and minute-scale fluctuations in nutrient availability as they swim through resource landscapes that are highly heterogeneous at sub-millimeter scales[1–5]. Surface-attached microorganisms experience rapid changes in resources through the movement of the liquid around them[6–8].

To understand the impacts that microorganisms have on the physiology of their hosts and on global elemental cycles, we have to understand how individual bacteria respond to nutrient fluctuations. However, our understanding of microbial physiology draws heavily on knowledge derived from steady environments or single transitions between steady states[9–11]. Microbial metabolism and growth under nutrient fluctuations remains a knowledge gap, largely due to the technical challenges of studying cells in highly dynamic environments. Here, we address this gap using single-cell growth experiments in a custom microfluidic device to show that rapid fluctuations substantially diminish growth, but also that bacteria can exhibit a fluctuation-adapted growth physiology that enhances growth under frequent environmental change.

Recent advances in single-cell measurement techniques have laid foundations for considering the implications of second- and minute-scale fluctuations on bacterial growth and physiology. Single-cell measurements of bacterial mass at femtogram resolution have confirmed that individual bacteria add mass exponentially[12,13]. Experiments enabled by a groundbreaking microfluidic tool, the Mother Machine, revealed that the exponential growth of individual cells is stable over hundreds of generations[14], indicating that steady-state growth applies not only at the population level but also to individuals. This seminal discovery has catalyzed major progress toward understanding the homeostatic regulation by which bacteria tune their growth and physiology to their environment, resulting, for example, in the Adder model of cell-size control[15–17] and hypotheses for its underlying mechanisms[18–20]. This wealth of literature stems from and reinforces a long-standing paradigm that each nutrient environment induces a characteristic steady-state growth rate[9,10,17] in which cells tightly regulate their size[17], proteome[21], and biosynthesis rates[10] in response to nutrient availability. The robustness of steady-state cell physiologies has led to growth laws that relate physiological traits, such as RNA–protein ratios[10,22], with steady-state growth rate.

The expansive experimental and theoretical characterization of steady-state growth has led to its use as the framework to interpret bacterial physiology and ecology, even in dynamic environments. Currently, our understanding of bacterial responses to changes in the environment derives heavily from characterizations of physiological transitions from one steady state to another. Upon a nutrient shift out of steady state, cells initiate a cascade of responses that depend on the nutrient composition of the new environment and require hours to complete[11,23]. Specific processes respond over a range of timescales—transcription over seconds, translation over minutes, cell division over hours[23]—and the progression of these physiological changes is reflected in a cell's growth rate. The dynamics of growth transitions thus provide insight into the strategies employed by bacteria and the ecological challenges under which these strategies have evolved[11,23–25].

Steady-state growth is highly informative when nutrients fluctuate on timescales longer than the timescales required for cells to transition between steady states; however, it is unclear whether it provides an appropriate framework for understanding physiology when nutrients fluctuate on timescales of seconds or minutes. For example, minute-scale alternation in the expression of metabolic pathways associated with different steady states may produce cells with proteins that are not usually co-expressed. Proteins expressed during prior exposures to a condition might reduce lag times when that condition returns[26,27], yet unnecessary gene expression can also reduce growth rate[21]. Understanding how frequent and repeated shifts in nutrient concentration integrate to affect bacterial growth physiology requires systematic study of single-cell growth under rapid nutrient fluctuations.

In this study, we characterized the rate and dynamics of bacterial growth under fluctuations between two fixed nutrient concentrations on timescales of seconds to minutes. Using a custom microfluidic device that precisely controls nutrient concentration over time, we quantified the growth dynamics of thousands of individual *Escherichia coli* cells exposed to identical, periodic nutrient fluctuations with periods as short as 30 s. We found that rapid nutrient fluctuations reduce growth rate by up to 50% when compared to a steady nutrient condition delivering the equal average concentration. However, the measured loss is considerably smaller than the growth loss expected from a null model based on measured growth responses to single shifts in nutrient concentration. Here, we provide the first evidence of a fluctuation-adapted growth physiology that alleviates growth loss in fluctuating nutrient environments and implicate continued temporal variability as a fundamental parameter for understanding bacterial physiology in dynamic habitats.

## Results and discussion

**Exposing single bacteria to precisely controlled, rapid nutrient fluctuations.** To determine how rapid nutrient fluctuations affect bacterial growth, we engineered a microfluidic device to rapidly switch between the delivery of two different nutrient concentrations while simultaneously imaging individual bacteria with time-lapse phase-contrast microscopy (Fig. 1a, "Methods", and Supplementary Fig. 1). We grew surface-attached *E. coli* under nutrient oscillations with periods of 30 s, 5 min, 15 min, or 60 min (Fig. 1b). Each experiment began by flowing cells from a growing batch culture into the device and allowing them to adhere onto its glass surface prior to initiating nutrient fluctuations ("Methods" and Supplementary Fig. 2). Switches between the two nutrient concentrations occurred while maintaining a constant flow rate and each switch was completed in <3 s (Supplementary Fig. 3). The resulting nutrient signal was reliably experienced by the cells as a square wave of equal time (half the period) in each concentration (Fig. 1b), with sharp transitions between concentrations (Fig. 1c). Neither nutrient depletion nor metabolite accumulation altered the composition of the nutrient media experienced by the cells, due to the high flow rates and channel depth (60 μm) used (Supplementary Fig. 4 and Supplementary Table 1). Usually, one of the two cells emerging from each cell division event was transported away by the flow (Fig. 1d), allowing us to acquire time series of thousands of individual cells (4000–20,000) for each experiment. We confirmed that growth rate was independent of a cell's position along the 10-mm-long region imaged within the microchannel (Supplementary Fig. 4) and that cells within the same condition therefore experienced identical nutrient signals over time.

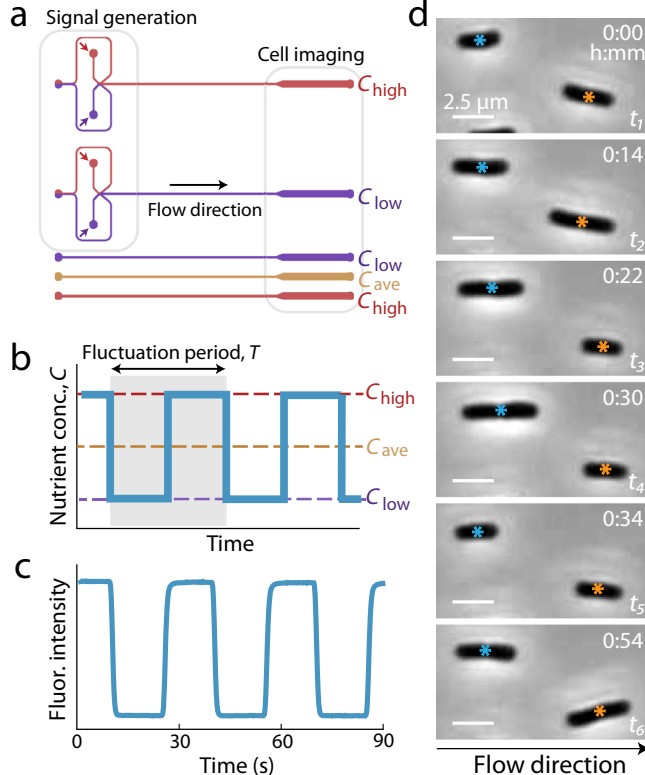

**Fig. 1 The microfluidic signal generator (MSG) creates automated, precise high-frequency fluctuations in nutrient concentration while enabling single-cell microscopy. a** The two channel configurations: the MSG switches between two media (top) and the straight channels each steadily deliver a single medium (bottom). Each experiment contains four parallel channels: one MSG and three straight. The two MSG channels displayed here schematically represent the flow conditions that deliver either $C_{low}$ or $C_{high}$ to the cells. The upstream portion of the MSG switches the nutrient media delivered to cells via automated control over the pressure differences driving each medium while maintaining a constant flow rate into the device. The top MSG delivers $C_{high}$ (red) to cells by pressurizing the red inlet higher than the $C_{low}$ (purple) inlet. The wider downstream section fits over ten imaging fields of view at ×60 magnification. **b** Bacteria were exposed to fluctuating signals in the form of even oscillations between a low and a high LB concentration ($C_{low}$ and $C_{high}$), with periods, $T$, between 30 s and 60 min. Three control environments, $C_{low}$, $C_{ave}$, and $C_{high}$, were run simultaneously with each fluctuating environment. **c** Fluorescence intensity, from fluorescein added to one of the media, illustrates the signal received at the cell imaging region over multiple oscillations ($T = 30$ s). Transitions between media are completed in <3 s. **d** Individual *E. coli* cells growing within the MSG. Cells were imaged at 117 s intervals; timestamps of selected images are displayed in minutes. Cells divide between $t_2$ and $t_3$ (orange) and between $t_4$ and $t_5$ (blue), and can be seen elongating between other frames. One of the two cells emerging from division is swept away with the flow. Images were cropped from one steady nutrient experiment and are representative of the growth behaviors observed in all experiments described in this paper.

To isolate the role of nutrient timescale from that of nutrient concentration, we switched between the same two nutrient concentrations for all fluctuating environments, a high and a low concentration of a complex growth medium ($C_{high} = 2\%$ LB, $C_{low} = 0.1\%$ LB), empirically chosen to avoid the saturation of growth rate (Supplementary Fig. 5a, b). Three control experiments with steady nutrient concentrations were run in parallel with each fluctuating experiment: one at $C_{high}$, one at $C_{low}$, and one at $C_{ave} = (C_{high} + C_{low})/2$ (i.e., 1.05% LB) (Fig. 1a). We found

that varying nutrient concentration reproduced key relationships between growth rate, cell size, and division time (Supplementary Fig. 5c, d) previously established by varying nutrient source (e.g., glucose vs. tryptic soy broth)[17], suggesting that variations in growth rate due to changes in nutrient concentration and nutrient source are physiologically similar. Importantly, the $C_{ave}$ control provided cells with average and total nutrient identical to that in the fluctuating environments. Together, these steady controls enabled us to distinguish the effects of fluctuation timescale from those of nutrient concentration by providing reference growth rates in steady low, average, and high nutrient concentrations.

**Growth rate rapidly responds to nutrient fluctuations.** This microfluidic system allowed us to measure the growth rate of individual cells in steady and fluctuating nutrient concentrations with high precision and temporal resolution. Growth rate, defined here as the rate at which cell volume doubles, was quantified from phase-contrast images of single cells acquired approximately every 2 min (see "Methods"). For each cell imaged, we extracted the length and width using image analysis and quantified cell volume, $V(t)$, by approximating the cell as a cylinder with hemispherical caps[18,28]. Using the resulting time series of cell volume (Fig. 2a), we computed the instantaneous growth rate for each single cell, $\mu(t)$, from $V(t + \Delta t) = V(t) \times 2^{\mu \Delta t}$ (see "Methods"). We focused on single-cell volumetric growth rather than cell replication or division time because biomass production has been shown to respond within minutes of a nutrient shift, whereas cell division responds more slowly (within an hour)[23]. The growth rate in the steady controls stabilized within 3 h of the start of the experiments (Supplementary Fig. 5). Thus, the steady-state growth rates ($G_{high}$, $G_{ave}$, and $G_{low}$) from steady $C_{high}$, $C_{ave}$, and $C_{low}$ were computed by averaging all single-cell growth rates, $\mu$, measured after 3 h within each respective condition (Fig. 2b). We confirmed with metabolomic profiling that the different steady-state growth rates between the three steady conditions resulted from changes in nutrient uptake rates, rather than changes in preferred metabolites (i.e., serine is likely the preferred metabolite across all conditions) ("Methods" and Supplementary Fig. 6).

Growth rate in the fluctuating environment changed rapidly in response to changes in nutrient concentration. Because instantaneous growth rate dynamics from single cells were noisy (Supplementary Fig. 7), we averaged the single-cell growth rates at each time point to visualize the dynamics of time-averaged instantaneous growth rates, which displayed strong and sharp fluctuations, changing more than two-fold within minutes (Fig. 2b). Instantaneous growth rate fluctuated with the immediate nutrient concentration, such that higher growth rates were observed in the high-nutrient phase and lower growth rates in the low-nutrient phase (Fig. 2b). While cell divisions occurred more frequently during phases of high nutrient, we confirmed that the fluctuations in growth rate reflected responses in single-cell volume, rather than cell division responses (Supplementary Fig. 8). When averaging the instantaneous growth rates from each phase of the nutrient signal, periodic changes were observed when nutrients fluctuated with a period of 5, 15, or 60 min (Fig. 2c), indicating that growth rate responded to nutrient shifts in <2.5 min. Changes in growth rate that slightly precede changes in the nutrient signal are not an anticipatory response, but rather are caused by limits in the time resolution of our measurements (Supplementary Fig. 9a). Similarly, fluctuations in growth rate were not resolvable with 30 s fluctuations, owing to the temporal resolution of image acquisition (2 min) (see Image acquisition in "Methods"). Nevertheless, these results establish that rapid

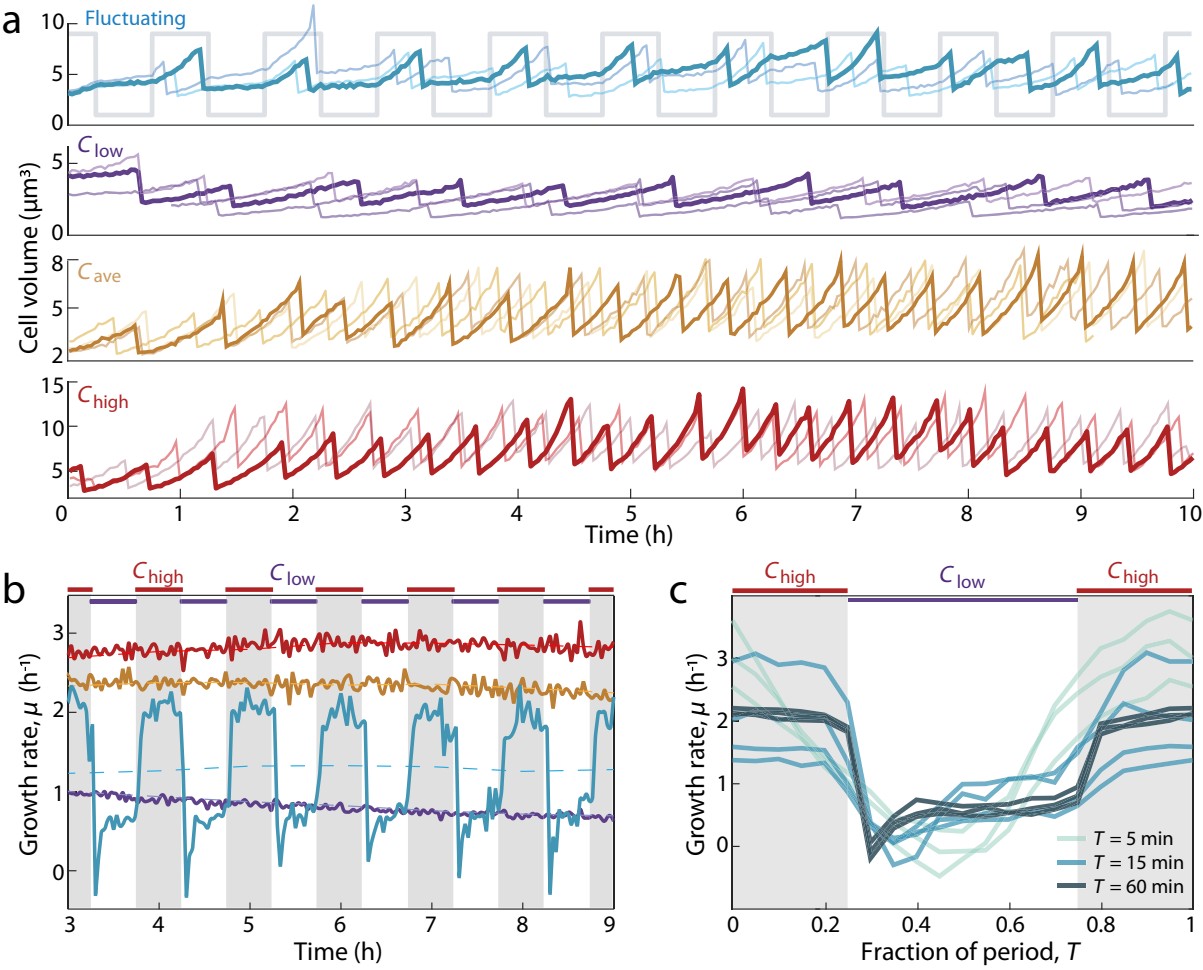

**Fig. 2 Nutrient upshifts and downshifts are followed by rapid adjustments in growth rate. a** Single-cell volume trajectories from a fluctuating environment (fluctuation period $T = 60$ min) and control environments: steady $C_{low}$, $C_{ave}$, and $C_{high}$. Each colored line tracks the repeated growth and division events of a single cell over several generations. One line is bold for clarity; the lighter lines are simultaneous tracks measured from different individuals. The gray line shows the nutrient signal in the fluctuating environment. In the fluctuating environment, single-cell volume grows at different rates depending on the nutrient phase. **b** Instantaneous growth rate, $\mu$, over time in steady and fluctuating environments. In the steady environments, $\mu$ is stabilized at steady-state growth by $t = 3$ h. In a fluctuating environment ($T = 60$ min), $\mu$ fluctuates with the nutrient signal. Each solid curve is a time-average of all instantaneous single-cell growth rates from each 2-min time bin, based on estimates of $\mu$ from at least 1842 total cells per replicate experiment. Each dashed curve is the time-average from each 30-min time bin. Given the noise in $\mu$ at the single-cell level (see Supplementary Fig. 7), the growth rate dynamics were best visualized by averaging many cells. **c** Instantaneous growth rate, $\mu$, averaged across all single-cell growth rates as a function of the nutrient phase in fluctuating environments. Regardless of the timescale of nutrient fluctuation ($T$), $\mu$ is higher in $C_{high}$ and lower in $C_{low}$. Regions shaded in gray correspond to $C_{high}$ phases of the fluctuating nutrient signal. Curves represent replicate experiments.

fluctuations in nutrient concentration lead to minute-scale fluctuations in instantaneous growth rate. We next asked how these nutrient fluctuations—and the resulting fluctuations in instantaneous growth rate—impact the mean rate at which individual cells grow.

**Rapid nutrient fluctuations reduce growth rate relative to steady conditions.** The mean growth rate in fluctuating environments, $G_{fluc}$, was consistently lower than the growth rate in the steady average conditions, $G_{ave}$. For each fluctuation timescale, we computed $G_{fluc}$ as the average of all instantaneous growth rates for all cells measured from 3 h to the end of the experiment. For nutrient fluctuations on 30 s, 5 min, 15 min, and 60 min periods, this measure yielded $G_{fluc}$ values of $1.93 \pm 0.16$, $1.53 \pm 0.20$, $1.15 \pm 0.28$, and $1.15 \pm 0.13$ h$^{-1}$, respectively (mean ± standard deviation; $n = 3$–4 replicate experiments per condition, each with at least 1842 cells; Fig. 3a and Supplementary Table 2). The

corresponding value of $G_{ave}$ was $2.31 \pm 0.18$ h$^{-1}$ ($n = 13$ replicate experiments; Supplementary Table 2). Accordingly, $G_{fluc}$ was lower than $G_{ave}$ by 16.5–50.2% of $G_{ave}$.

This reduction in mean growth rate has strong implications for bacterial population dynamics. For example, for each initial biomass of $M_0 = 1$ μm$^3$, the measured $G_{fluc}$ from 30 s fluctuations ($1.93 \pm 0.16$ h$^{-1}$) and $G_{ave}$ ($2.31 \pm 0.18$ h$^{-1}$) correspond to a daily ($t = 1$ day) produced biomass of $9 \times 10^{13}$ and $5 \times 10^{16}$ μm$^3$, respectively (Supplementary Table 3). In other words, two cells of equal initial volume, both growing exponentially ($M(t) = M_0 \cdot 2^{Gt}$) with one at rate $G_{fluc}$ and one at $G_{ave}$, will differ in biomass production by >500-fold in a single day. Thus, by affecting growth rate, the timing of nutrient availability is an important parameter in the growth and productivity of bacterial populations.

Why do nutrient fluctuations reduce growth rate relative to the steady-state growth rate, $G_{ave}$? A simple mathematical model, relevant to the concave Monod curve, illustrates a physiological implication of fluctuating nutrients. In the Monod curve, steady-

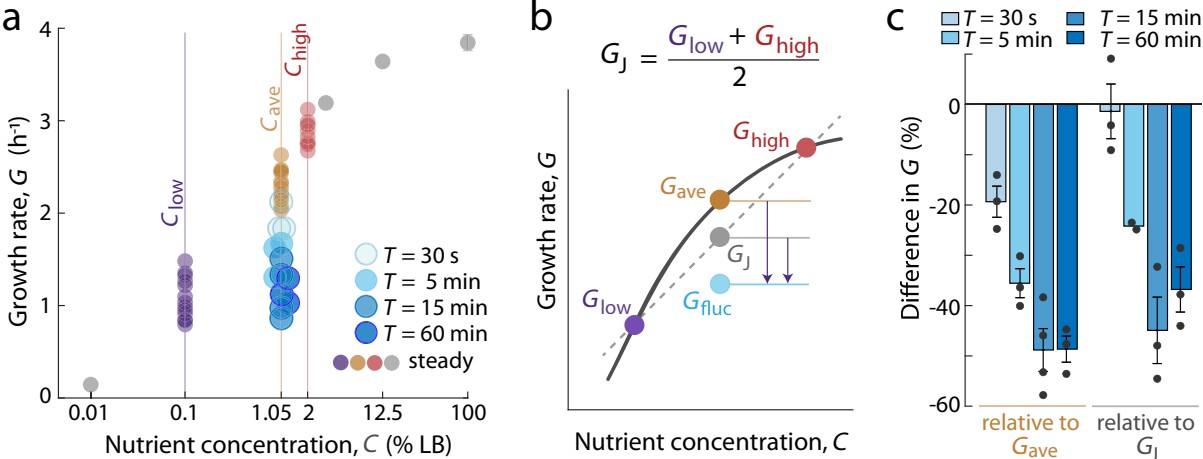

**Fig. 3 Rapid nutrient fluctuations reduce growth rate compared to environments of equal average nutrient concentration. a** Cells in fluctuating environments of various period lengths ($T$) experienced the same time-averaged nutrient concentration as $C_{ave}$ but grew at lower growth rates. Each point represents the mean growth rate of all individual cells measured from all time steps after the initial 3 h of each experiment. Colored points are replicate experiments for steady $C_{low}$, $C_{ave}$, or $C_{high}$ or fluctuating nutrient conditions; gray points are additional steady nutrient concentrations that span the range from nearly zero growth to saturated growth. Error bars denote the standard error of the mean and are smaller than data points when not visible. **b** Schematic representation of Jensen's inequality ($G_{ave} > G_J$) and the relationship between $G_{ave}$, the growth rate at steady nutrient $C_{ave} = (C_{low} + C_{high})/2$, and $G_J$, the maximum growth rate expected for cells spending equal time at $C_{low}$ and $C_{high}$. $G_{fluc}$ in our experiments is below $G_J$. **c** Percent difference in growth rate for $G_{fluc}$ at different period lengths relative to a steady-state reference, either $G_{ave}$ or $G_J$. Percent difference was calculated from $n = 2$–4 biologically independent experiments with conditions performed on the same day as $(G_{fluc} - G_{ave})/G_{ave} \times 100$ or $(G_{fluc} - G_J)/G_J \times 100$. The raw values used in these calculations are reported in Supplementary Table 6. Data points indicate the difference calculated from each biological replicate. Error bars denote the standard error of the mean.

state growth rate increases less than linearly with nutrient concentration (Fig. 3b). This steady-state function is an example of Jensen's inequality, which states that for a concave function, the mean of the function (i.e., $G_J = (G_{low} + G_{high})/2$) is smaller than the function of the mean (i.e., $G_{ave}$). This inequality predicts that fluctuations between $C_{high}$ and $C_{low}$ would result in a growth rate, $G_J$, lower than $G_{ave}$ ($G_J = 1.97 \pm 0.16\,h^{-1} < G_{ave} = 2.31 \pm 0.18\,h^{-1}$) (Fig. 3b). Still, Jensen's inequality does not explain the observed growth reductions. The growth rates from fluctuating environments, $G_{fluc}$, were lower than $G_J$ for all fluctuation periods except 30 s (Fig. 3c). This difference is consistent with the unrealistic scenario represented by Jensen's inequality, which considers a cell that fluctuates between growth at two steady states, $G_{low}$ and $G_{high}$ ($1.07 \pm 0.23$ and $2.86 \pm 0.14\,h^{-1}$, respectively). In reality, the magnitude by which $G_{fluc}$ is lower than $G_J$ depends on the dynamics by which single-cell growth rate responds to fluctuations in nutrient concentration.

We hypothesized that the reduction in growth rate results from the time required for cells to adopt the steady-state physiology characteristic of the current nutrient condition after each fluctuation. The prevailing paradigm for growth transitions presumes that cells initiate a physiological transition to the immediate nutrient environment[11], regardless of the nutrient timescale. In response to a single shift in nutrient concentration (e.g., from $C_{low}$ to $C_{high}$), cells grow at rates lower than steady-state $G_{high}$ for several hours until the physiological transition is complete (Fig. 4a). This hours-scale transition in growth rate is a characteristic response to environments in which the nutrient condition shifts only once[11,23–25]. When nutrient fluctuations occur on timescales longer than this hours-scale transition in growth rate, predicting growth dynamics from such single-shift responses is relatively straightforward[29]. When nutrient fluctuations occur on timescales faster than the time required to physiologically transition between steady states, the paradigm predicts that cells should never stabilize in growth rate and that

cells continuously grow at rates lower than steady state, causing $G_{fluc}$ to be lower than $G_J$.

**Growth rate responses differ between repeated fluctuations and single nutrient shifts.** To determine whether the timescale of physiological transitions could explain the reduction in $G_{fluc}$, we compared the growth rate dynamics between cells exposed to fluctuations and cells exposed to a single up- or downshift in nutrient concentration after having reached steady state prior to the shift. In single-upshift experiments, cells growing steadily at $G_{low}$ were switched to $C_{high}$, while in single-downshift experiments, cells growing steadily at $G_{high}$ were switched to $C_{low}$. These single-shift experiments enabled us to quantify the time required for cells to physiologically transition between the two steady states, as well as the dynamics of growth rate across the transition (Fig. 4a). In single-upshift experiments, the growth rate gradually increased from $G_{low}$ until reaching steady-state $G_{high}$ after 2–3 h. In single-downshift experiments, the growth rate dropped sharply from $G_{high}$ down to 10% of $G_{low}$, then gradually increased until reaching steady-state $G_{low}$ after 5 h (Fig. 4a).

In contrast to cells exposed to single shifts in nutrient concentration, cells grown in fluctuations stabilized at growth rates lower than steady state. We observed this stabilization of growth rate in fluctuating environments with 15 and 60 min periods, which were better resolved with our 2-min imaging interval (Fig. 4b). Growth rate under these fluctuations stabilized at 66% of $G_{high}$ after each upshift and at 63–68% of $G_{low}$ after each downshift (Fig. 4c, d and Supplementary Table 4) within minutes: $3.8 \pm 0.0$ or $3.3 \pm 1.4$ min after each upshift and $2.2 \pm 1.9$ min or $15.0 \pm 7.6$ min after each downshift ($T = 15$ or 60 min, respectively) (Fig. 4e). The minute-scale stabilization of growth rate observed from fluctuating conditions was in stark contrast to the hours-long timescale observed from single-shift conditions ($116.3 \pm 12.4$ min after a single upshift and at least 297.5 min after

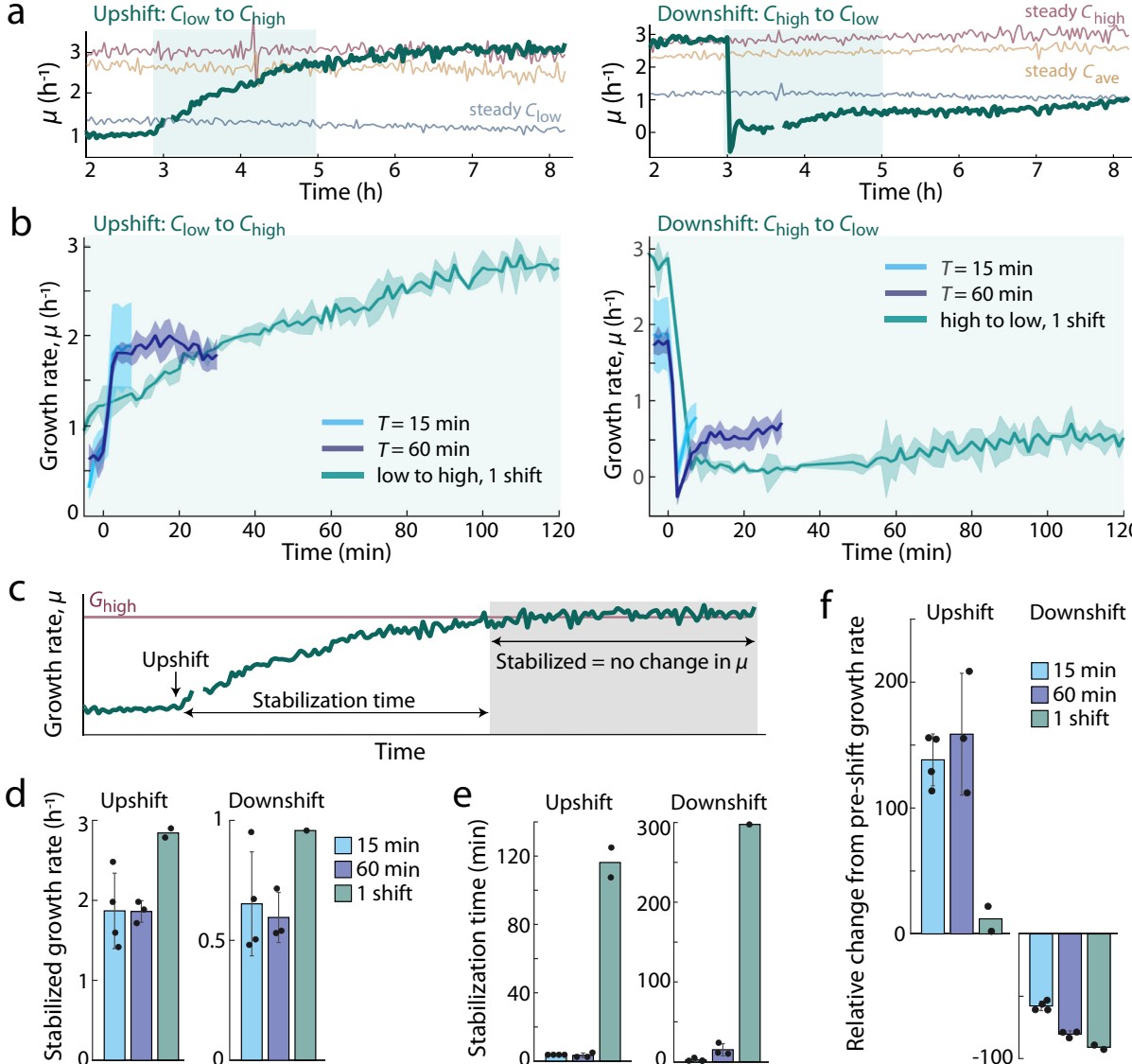

**Fig. 4 Growth rate responses differ between repeated fluctuations and single nutrient shifts. a** Average growth rate of single cells over time in four conditions: a single shift in nutrient concentration (occurring at 3 h, after cells had reached steady-state growth in the initial condition), steady $C_{low}$, steady $C_{ave}$ and steady $C_{high}$. On the left, a single nutrient upshift (shift from $C_{low}$ to $C_{high}$), and on the right, a single nutrient downshift (from $C_{high}$ to $C_{low}$). After each shift, the growth rate gradually reaches steady-state growth in the post-shift condition. The growth rates in $C_{low}$ before the upshift and in the steady $C_{low}$ condition are both within the range of measured steady-state $G_{low}$ (Supplementary Tables 2 and 7). Data are from one representative experiment. The shaded region marks the portion of single-shift data plotted in (**b**), which does not include the full progression to steady-state growth after a single up- or downshift. **b** Growth rate of single cells in fluctuating nutrient conditions stabilizes more rapidly and at a lower value when compared to the growth rate dynamics of cells experiencing a single shift. Data were aligned such that the nutrient shift in all conditions occurs at $t = 0$. Post-shift data in fluctuating environments are plotted up until the next shift occurs. Shaded error bars denote the standard deviation of the mean among replicate experiments ($n = 3$–4 for fluctuating conditions; $n = 2$ for single-shift conditions). **c** Growth rate is considered stabilized once the slope of the growth signal within a shrinking window reaches zero. Stabilization time is defined as the time between the nutrient shift and the time at which the growth rate is stabilized. **d** The growth rate of fluctuation-grown cells stabilized at rates lower than steady-state $G_{high}$ or $G_{low}$. Cells experiencing 15 and 60 min fluctuations stabilized at $1.86 \pm 0.47$ and $1.86 \pm 0.13\ h^{-1}$, respectively, after an upshift and at $0.65 \pm 0.22$ and $0.60 \pm 0.10\ h^{-1}$ after a downshift. Cells shifted once from steady-state stabilized only upon reaching steady state $G_{high}$ ($2.84 \pm 0.08\ h^{-1}$) after an upshift or $G_{low}$ ($0.96\ h^{-1}$) when growth rate stabilized after a downshift (only one of two replicates stabilized at $G_{low}$ after 5 h of post-shift observation). **e** Cells grown in fluctuations stabilize in growth rate within $3.8 \pm 0.0$ ($T = 15$ min) or $3.3 \pm 1.4$ min ($T = 60$ min) of each upshift and within $2.2 \pm 1.9$ ($T = 15$ min) or $15.0 \pm 7.6$ min ($T = 60$ min) of each downshift ($n = 3$–4). Cells grown in steady environments stabilize hours after a single shift, $116.3 \pm 12.4$ min in the case of upshifts ($n = 2$) and at least $297.5$ min after a downshift (one of two replicates stabilized after 5 h). **f** The initial change in growth rate in the minutes following a nutrient shift differed between cells experiencing fluctuations and cells experiencing only a single shift. Over the first 7.5 min after each upshift, cells experiencing fluctuations increased in growth rate relative to their pre-shift growth rate ($t = 0$) by $138.3 \pm 20.6\%$ (15 min) or $158.7 \pm 48.4\%$ (60 min), compared to a $12.0 \pm 14.0\%$ increase in the single-shift case. Over the 7.5 min after each downshift, cells experiencing fluctuations decreased in growth rate by $57.8 \pm 3.7\%$ (15 min) or $80.4 \pm 2.6\%$ (60 min), compared to a $90.9 \pm 2.5\%$ decrease in growth rate in cells after a single shift. **d**-**f** Error bars denote the standard deviation of the mean of $n = 3$ or 4 biologically independent experimental replicates (none displayed for $n < 3$). Overlaid data points represent measurements from each of $n$ replicates.

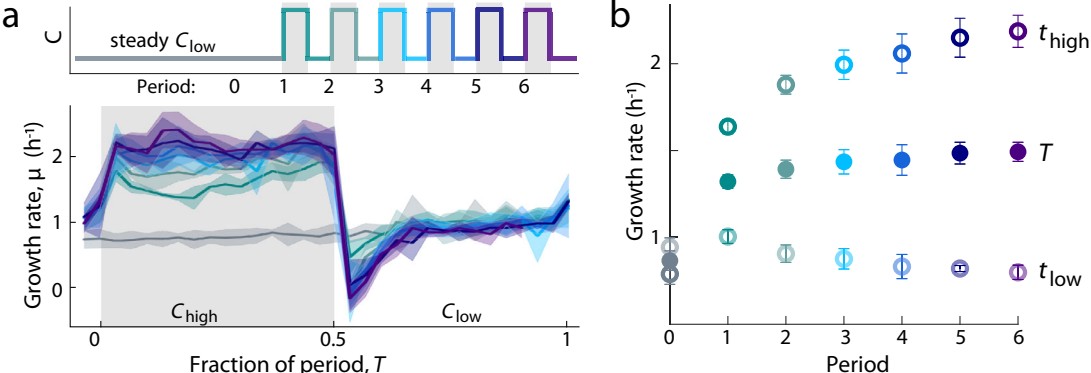

**Fig. 5 Rapid nutrient fluctuations induce the transition to a fluctuation-adapted growth physiology. a** After at least 3.5 h of growth in steady $C_{low}$, cells experience the onset of nutrient fluctuations ($T = 60$ min) between $C_{low}$ and $C_{high}$. The growth rate dynamics in response to each successive nutrient period transition from the dynamics observed in response to a single shift to those observed in fluctuating environments (Fig. 4b). The growth rate dynamics of the successive periods, including the 60 min preceding the first nutrient upshift (period 0), are overlaid to show the stabilization of the growth rate signal. Shaded error bars denote the standard deviation of the mean among replicate experiments ($n = 3$). **b** Stabilization of growth rate dynamics with successive nutrient periods. The mean growth rate following each downshift (half-period in $C_{low}$, $t_{low}$), each upshift (half-period in $C_{high}$, $t_{high}$) and across the full period ($T$) adjusts and stabilizes by the third period. Error bars denote the standard error of the mean ($n = 3$) and are smaller than points when not visible.

a single downshift) (Fig. 4e) and is more consistent with a stable growth physiology, i.e., a single cellular composition that grows at different rates due to changes in metabolite flux (Fig. 4f), than changes in gene expression.

These observations show that the effect of rapid nutrient fluctuations on growth cannot be understood by reference to a sequence of single up- and downshifts. Instead, the minute-scale stabilization of growth rate and distinct values of growth rate (once stabilized) suggests that cells grown under fluctuations have a different growth physiology from cells grown under steady nutrient conditions. While we measured different stabilization times between the two fluctuating timescales after downshifts, the overall growth rate dynamics observed from both conditions ($T = 15$ and 60 min) were more comparable with each other than with the single downshift condition. Growth rate dropped sharply after a shift to $C_{low}$ in fluctuating and single-shift conditions (Fig. 4b), yet growth rate quickly increased in both fluctuating conditions while no such rebound occurred after a single shift (Fig. 4b).

That cells grown in $T = 15$ and 60 min nutrient fluctuations induced growth rate responses, distinct from cells responding to single shifts, suggests the existence of a distinct physiology adopted by cells upon experiencing rapid nutrient fluctuations. Growth rate stabilized at comparable values in both fluctuating timescales (Fig. 4d), suggesting that $T = 15$ and 60 min nutrient fluctuations induce the same or very similar physiologies. Future work to access the proteomic and transcriptomic responses to fluctuations will be important to understand the molecular mechanisms of adaptation to fluctuations. Currently, our observations do not support the hypothesis that the reduction in growth rate under fluctuations results from continued physiological transitions toward growth at $G_{high}$ and $G_{low}$. Instead, they led us to hypothesize that rapid fluctuations may induce a fluctuation-adapted physiology, one that remains stable across minute-scale changes in nutrient concentration.

**Rapid nutrient fluctuations induce a fluctuation-adapted growth physiology.** To demonstrate that bacteria can adopt a distinct physiological state when exposed to rapid nutrient fluctuations, we performed an experiment in which cells growing under steady $C_{low}$ for at least 3.5 h were then exposed to

fluctuations of $T = 60$ min. We found that the first nutrient upshift induced the gradual increase in growth rate characteristic of the physiological transition between $G_{low}$ and $G_{high}$ observed from single-shift conditions (Fig. 5a). Subsequent upshifts displayed faster growth rate adjustments that increasingly resembled that characteristic of cells grown in fluctuating conditions (Fig. 5a), confirming that cells can adopt a fluctuation-induced growth physiology, induced by repeated nutrient shifts.

The transition to a stable growth physiology occurred within 2–3 h of the onset of fluctuations. Growth from each successive nutrient upshift (growth in $C_{high}$) increased during this 2–3 h transition time, offsetting the decreasing growth following each successive downshift (growth in $C_{low}$) (Fig. 5b), and ceased to differ significantly from the third period of nutrient fluctuation onwards (Fig. 5b). The overall increase in growth between the initial and stabilized periods suggests that adopting the fluctuation-induced physiology enhances growth in fluctuating environments. This enhancement is potentially a physiological trade-off that limits the maximum growth rate in any given nutrient concentration to increase the cell's potential for growth when higher nutrient concentrations become available (Fig. 5b). Indeed, in the minutes after a nutrient shift, growth rate measured from cells in fluctuating conditions was higher than that of cells exposed to a single shift (Fig. 4f and Supplementary Fig. 9b). Thus, the stable physiology adopted by cells exposed to rapid nutrient fluctuations can alleviate the potential reduction of growth by unsteady conditions.

**Growth in rapid nutrient fluctuations is higher than predicted by single-shift growth models.** To quantify the advantages of a fluctuation-induced physiology, we compared mean growth rates under fluctuations with mean growth rates predicted by a null model. We constructed the null model from the growth rate dynamics quantified from single shifts (Fig. 4a), simulating growth rate over time under nutrient fluctuations while assuming single-shift dynamics (Fig. 6a). Two different assumptions produced different dynamics under minute-scale fluctuations, yet predicted the same trends (Supplementary Fig. 10 and Construction of Null Model in the Supplementary information). These dynamics were then averaged to predict $G_{fluc}$ in the absence of a fluctuation-induced physiology across a range of fluctuation

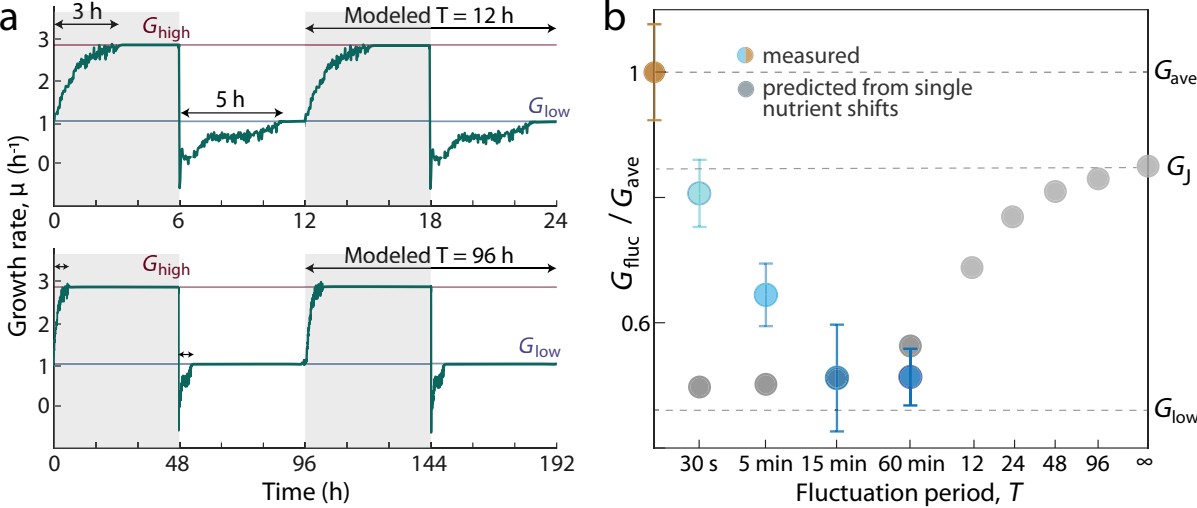

**Fig. 6 Growth rate under rapid nutrient fluctuations is sensitive to timescale and higher than predicted by a null model based on single shifts. a** Instantaneous growth rate dynamics modeled using growth rate responses measured from single nutrient shifts. The dynamics modeled for two slow nutrient timescales ($T = 12$ and $96$ h) are illustrated here. At these timescales, each phase of $C_{high}$ or $C_{low}$ is substantially longer than the 3 or 5 h required for cells to reach either steady-state $G_{high}$ or $G_{low}$ after a single up- or downshift, respectively. The models used for growth rate dynamics at faster fluctuation timescales are fully described in Supplementary Fig. 10a, b and Construction of Null Model in the Supplementary information. For each modeled timescale, the growth rate dynamics were time-averaged to calculate the predicted average growth rates, $G_{fluc}$, plotted in (**b**). **b** Growth rate under rapid nutrient fluctuation ($G_{fluc}$, blue) is higher than predicted from data on single nutrient shifts between $C_{high}$ and $C_{low}$. The plot shows $G_{fluc}$ as a fraction of the growth rate in the steady average nutrient environment ($G_{ave}$; gold). Each measured point represents the time-averaged growth rate $G$ and standard deviation of the mean among replicates ($n = 3$–$4$). Predicted values of $G_{fluc}$ (gray) reach a maximum of $G_J$ when the fluctuating nutrient timescale is infinitely long relative to the time required to transition from growth at $G_{low}$ to $G_{high}$, and vice versa. The deviation between measured and predicted $G_{fluc}$ differentiates the growth physiology at rapid fluctuation timescales from the growth behaviors expected from single shifts, and highlights the timescale-dependent nature of the growth advantage conferred by a physiology adapted for growth under rapid nutrient fluctuations.

timescales ($T = 30$ s to $96$ h) (Fig. 6b and Supplementary Table 5). For very slow fluctuations, the null model predicted that $G_{fluc}$ approaches the value predicted by Jensen's inequality, $G_J$ (Fig. 6b), as cells spend almost the entirety of each nutrient phase at either $G_{low}$ or $G_{high}$ (Fig. 6a). Across all nutrient timescales $T$, the null model predicted that $G_{fluc}$ diminishes with decreasing $T$ (Fig. 6b).

This trend in the null model is the opposite of that displayed by the measured values of $G_{fluc}$ for rapid fluctuations. Experimentally, we measured an increase in $G_{fluc}$ with decreasing fluctuation period: $G_{fluc}$ is 50% below $G_{ave}$ for $T = 60$ min, but only 16.5% below $G_{ave}$ for $T = 30$ s (Fig. 6b). Relative to $G_{ave}$, the measured $G_{fluc}$ represents only $70.8 \pm 10.0\%$ of the predicted growth lost for $T = 5$ min and $38.9 \pm 10.8\%$ for $T = 30$ s (mean and standard deviation, $n = 3$). This increase in measured $G_{fluc}$ over the null model predictions represents the growth advantage in fluctuating conditions afforded by the fluctuation-induced growth physiology. These results demonstrate that growth rate in rapid fluctuations is not only quantitatively distinct from the values of $G_{fluc}$ predicted from single-shift dynamics, but also shows a qualitatively different trend in $G_{fluc}$ across nutrient timescales. While the values of $G_{fluc}$ predicted by the null model decreased with decreasing nutrient timescale, measured $G_{fluc}$ increased with decreasing nutrient timescale, suggesting that the cellular physiology induced by rapid fluctuations is an adaptation suited to alleviate some of the growth rate losses imposed specifically by second- and minute-scale nutrient fluctuations.

A remaining question concerns how cells sense that the environment is rapidly fluctuating and initiate the transition to a fluctuation-induced growth state. It is unclear how the fluctuation-induced physiology is achieved, in part because we do not know how gene expression or function may differ between cells growing

in fluctuations and cells growing at steady state; however, prior work in bulk chemostat systems delivering transient minutes-long pulses of glucose may offer important insights[30–32]. For example, in glucose-limited conditions, some yeast species (including *Saccharomyces cerevisiae*) respond to a pulse of glucose by activating alcoholic fermentation, while others prevent ethanol production by increasing acetaldehyde oxidation[33]. Yeasts that ferment the pulsed glucose into ethanol instead of biomass generally exhibit no change in growth rate across the pulse, whereas yeasts avoiding pulse-activated fermentation exhibit increased biomass production upon the addition of glucose[33]. The dependence of growth rate in dynamic environments on the allocation of metabolic flux has also been proposed to occur in *E. coli*, which uses the carbon storage molecule glycogen to maintain higher growth rates under pulsing nutrient conditions[34]. Experiments to test whether similar processes may explain our findings could be performed by adapting such bulk systems to measure gene expression and metabolic activity under minute-scale nutrient fluctuations.

Alterations in gene expression can be beneficial when environments change on rapid timescales. For example, increased expression of photoprotection proteins has been demonstrated to increase the growth yield of plants exposed to minute-scale fluctuations in light[35]. Alternatively, post-translational activation of so-called spare ribosomes has been implicated as a bacterial strategy to increase growth rate rapidly (i.e., without new protein synthesis) upon nutrient upshift[24,25,30]. *E. coli*, upon sensing depleted intracellular amino acid levels, has also been observed to induce broad responses to increase energy production, ribosome levels, and translational capacity[36]. These observations illustrate the diversity of strategies that life has evolved to cope with the challenges of environmental change. By reporting a fluctuation-adapted physiology with growth benefits

in rapidly fluctuating environments, we contribute a framework by which to pursue an understanding of bacterial growth that is relevant to realistic habitats.

## Accounting for fluctuations in models of bacterial growth.

Our study is the first to observe single-cell responses to rapid nutrient fluctuations and report a fluctuation-induced growth physiology. We found that bacteria exposed to nutrient fluctuations exhibit reduced growth rates compared to that of the steady average nutrient condition, even when nutrients fluctuated on timescales as rapid as seconds. These reductions are not explained by the current paradigm in bacterial physiology, which holds that cells experiencing a shift in their nutrient environment will begin the transition to the steady-state growth physiology of the post-shift environment[11,23–25]. Despite the evidence that bacteria encounter rapid and repeated nutrient fluctuations in their environments[1,2,4,5,8,37–41], single shifts in nutrient composition remain the dominant method by which bacterial growth is studied in dynamic nutrient conditions[11,24,25]. This study reports key differences in growth rate dynamics between fluctuations and single shifts, thus introducing repeated fluctuation as a better experimental system to study variability in microbial habitats than the classic single-shift paradigm.

In finding starkly different responses between fluctuations and single shifts, our study presents a conceptual advance important for the interpretation of past work and the direction of future work considering microbial growth in complex environments. By quantifying single-cell growth rate across a range of fluctuation timescales relevant to many bacterial habitats, this work demonstrates that growth responses to fluctuations are not simply transitions between two discrete states. Instead, we uncovered a strong dependence of bacterial growth on the temporal dynamics of nutrient concentration, highlighting the importance of temporal variability (and by extension, microscale heterogeneity) when considering bacterial growth in realistic environments. This work establishes nutrient timescale as a fundamental parameter characterizing bacterial environments, forming a third axis (in addition to chemical composition and temperature) to consider when studying bacterial growth.

The discovery of a fluctuation-induced growth physiology in *E. coli* highlights the importance of temporal context in bacterial regulation of growth with nutrient availability. Further studies with diverse patterns of nutrient fluctuations may yield additional strategies of bacterial growth in temporally variable environments. It is also possible that distinct strategies of growth in complex temporal environments have evolved between bacterial species that occupy distinct ecological niches. Taxon-specific specializations for the timescale of light fluctuations have long been observed in plants[42], and more recently in microbial response times to inputs of water to dry soils[43].

Our results illustrate how identical environmental shifts can induce different responses depending on the timescale at which the shifts are delivered, demonstrating the need to account for temporal variability in the environment at timescales that have been mostly ignored to date. Understanding the diversity of growth responses to realistic features of microbial environments will bring us closer to the establishment of general frameworks for bacterial growth in natural ecosystems and the discovery of mechanistic links between the interactions that occur on the scale of single cells, populations, and communities.

## Methods

**Bacterial strain**. All experiments in this study were performed with the same *E. coli* strain, K-12 NCM3722 Δ*motA*. The motility mutant (Δ*motA*) lacks flagella, facilitating long-term observation in microfluidics[17], and is derived from the

background strain K-12 NCM3722, which lacks the growth defects observed in other strains of *E. coli*[44,45].

### Growth media

*Batch culture medium.* MOPS (3-(*N*-morpholino)propanesulfonic acid) medium (Teknova) supplemented with 0.2% glucose w/v and 1.32 mM K$_2$HPO$_4$ was used for overnight and seed batch cultures. All batch cultures were 3 mL of supplemented MOPS medium inoculated with *E. coli*.

*Microfluidics medium.* Lysogeny broth (LB) composed of tryptone (10 g/L), yeast extract (5 g/L), and NaCl (10 g/L) was used for all microfluidic nutrient conditions. For the microfluidic experiments, the same stock solution of 100% LB was mixed with an equimolar NaCl solution (2.5 g NaCl in 250 mL 0.22 μm filtered water; Millipore Millipak Express 40, catalog no. MPGP04001) to prepare three dilutions: low, average, and high. To avoid bubble formation within the microfluidics, the NaCl solution was freshly autoclaved the day of each experiment and then cooled before preparing the LB dilutions. The high LB (2%) mixed 2 mL of the full LB into 98 mL salt solution. The low LB (0.1%) mixed 5 mL of the high LB solution with 95 mL salt solution. Afterward, the high LB solution was labeled with 0.26 nM sodium fluorescein, to allow visual calibration of switching between media. All solutions were adjusted to pH 7 with NaOH. Equal parts of low and high LB were mixed to produce the average LB control; hence, the average LB medium contained 0.13 nM sodium fluorescein. This fluorescein addition had no effect on the growth rate (Supplementary Fig. 2a). Furthermore, we confirmed with metabolomic profiling that the different steady-state growth rates between the three growth conditions resulted from concentration-dependent changes in nutrient uptake rates, rather than changes in preferential metabolite uptake (Supplementary Fig. 6). Growth media were loaded into plastic 10 mL syringes (Codan) or glass vials (VWR, cat. no. 548-0154) and warmed to 37 °C at least 3 h prior to the start of each experiment.

**Metabolomics characterization of growth media**. Changes in extracellular nutrient concentrations can affect uptake rates (due to variations in transporter affinity) or uptake order (in rich media, *E. coli* have been observed to deplete preferred metabolites before beginning to consume others[46]). To determine whether the preferred nutrient sources in our microfluidics experiments differed with nutrient concentration, we measured the depletion of extracellular metabolites from batch cultures. Batch cultures were necessary to observe metabolite depletion by enabling higher cell densities per volume. Measurements of depletion from the earliest time points after inoculating the batch cultures (i.e., when the media composition was least changed) best represented the nutrient consumption in microfluidic conditions, which continuously replenished all metabolites.

We collected the supernatant from 20 mL batch cultures grown at 37 °C with shaking in 125 mL Erlenmeyer flasks. Twelve flasks were prepared in parallel, four of each nutrient concentration ($C_{low}$, $C_{ave}$, and $C_{high}$). Three of each concentration were inoculated with cells, while the fourth flask was kept bacteria-free and sampled as a blank control. The inoculum was prepared with the same overnight and seed culture preparation used for the microfluidics experiments (see Cell preparation). Once the seed culture reached an OD$_{600}$ of 0.1 (grown in MOPS medium with 0.2% glucose), nine 1 mL aliquots were centrifuged for 2 min at 2500 r.c.f. (Eppendorf, Centrifuge 5424R) and the MOPS-based supernatant was removed. The cell pellet was gently resuspended in the final growth medium ($C_{low}$, $C_{ave}$, or $C_{high}$) and then inoculated into the appropriate flask. Each flask was sampled every 30 min by centrifuging 500 μL of culture for 5 min at 2500 r.c.f. Then, 100 μL of supernatant was removed from the top of each tube and stored in a 96-well plate (Thermo Scientific). Samples were kept on ice when in 1.5 mL microcentrifuge tubes (Sarstedt AG & Co.), and then at −20 °C when in the plate. The samples were thawed and diluted 1:10 in milliQ water prior to direct injection and were measured with flow-injection time-of-flight mass spectrometry.

Untargeted metabolomics measurements were performed with a binary LC pump (Agilent Technologies) and an MPS2 Autosampler (Gerstel) coupled to an Agilent 6520 time-of-flight mass spectrometer (Agilent Technologies) operated in negative mode, at 4 Ghz, high resolution, with an *m/z* (mass/charge) range of 50–1000[47]. The mobile phase consisted of isopropanol:water (60:40, v/v) with 5 mM ammonium fluoride buffer at pH 9 at a flow rate of 150 μl/min. Raw data were processed and analyzed with preprocessing raw mass spectrometry data functions contained in the bioinformatics toolbox of Matlab (The Mathworks, Natick)[47]. Overall, we detected 7037 ions, of which 284 could be annotated against the KEGG database (restricted to *E. coli*) with 0.003 Da tolerance. We monitored the depletion of these 284 metabolites over time in batch cultures inoculated into $C_{ave}$ or $C_{high}$. We found that detectable metabolites displayed comparable dynamics in the two nutrient concentrations (Supplementary Fig. 6). This suggests that the different growth rates observed among nutrient concentrations arise from differences in nutrient flux and not from differences in the composition of nutrient consumed. Metabolites in $C_{low}$ were below the detection limit.

**Cell preparation**. Cells for each experiment were grown in two batch cultures, the overnight culture and the seed culture, before entering the microchannels. The overnight culture was inoculated directly from a −80 °C glycerol stock into 3 mL of supplemented MOPS medium and shaken for 12–16 h at 37 °C at 200 r.p.m. The

next morning, cells from the overnight culture were diluted to achieve 3 mL of supplemented MOPS medium with an initial $OD_{600}$ below detection, generally a 1:1000 or 1:2000 dilution. This seed culture was used to inoculate microchannels once cells reached an $OD_{600}$ between 0.07 and 0.10.

**Microchannel fabrication**. Microfluidic channels with a depth of 60 µm were cast in polydimethylsiloxane (PDMS) from a custom-made master mold such that all four channels (one microfluidic signal generator (MSG) for fluctuating environments and three straight channels for steady environments) were present on the same device (Fig. 1a). Each PDMS device was bonded to a glass slide by plasma treating each interacting surface for at least 1 min, then incubating the assembled chip for at least 2 h at 80 °C. The morning of each experiment, bonded channels were cooled to room temperature and then treated with a 1:10 dilution of poly-L-lysine (Sigma, catalog no. P8920) in milliQ water. Poly-L-lysine treatment increased the number of attached cells and extended attachment duration, allowing for longer observations of single cells without affecting growth (Supplementary Fig. 2b, c). This treatment has no effect on the growth rate (Supplementary Fig. 2c) and involves incubating the diluted poly-lysine solution inside each channel for 15 min, before gently removing the solution and flushing the emptied channel with sterile milliQ water. The treated channels were then air-dried for at least 2 h prior to experimental use.

**Nutrient signal calibration and generation**. All fluctuating nutrient signals in this study switched between two nutrient media: a high and low concentration of LB. To switch between media, we oscillated pressure within each reservoir of the nutrient medium while maintaining a steady mean pressure to ensure a steady total flow rate of the medium through the device (Supplementary Fig. 1). This flow rate was determined by collecting the fluid output from the MSG and measuring the volume per minute. While the pressure differentials across the setup can vary (e.g., different device, slight variations in tubing lengths and angles), the range of flow rates used had no effect on growth rate (Supplementary Fig. 2). Because the pressure differentials could vary from day to day, the pressure differences required to completely switch between media were calibrated prior to each experiment at ×20 magnification, which enabled visualization of the entire signal junction (Supplementary Fig. 1). These calibrated pressure differences were then used to define the fluctuating nutrient signal. The pressure system was programmed to generate the signal in synchrony with image acquisition. Thus, timestamps from the image data could be directly correlated with specific time points within the nutrient signal. Separately, the stability of the calibrated signals was visually confirmed by comparing the fluorescent signal exiting the junction with the signal observed downstream (Fig. 1c and Supplementary Fig. 3). These visualizations were conducted at ×60 magnification and quantified by custom image processing scripts in MATLAB (see Code availability).

**Quantification of switching timescale and nutrient signal stability**. To assess the correspondence between our designed signal (an even square-wave) and the signal realized within the microfluidic device (Fig. 1c), we compared the transition dynamics of nutrient shifts occurring immediately after the signal junction and those occurring near the end of the cell imaging region. Two positions in the MSG were imaged while the fluid flowing through fluctuated between a medium labeled with 0.26 nM sodium fluorescein and an unlabeled medium on a 30 s period. Specifically, we compared the sharpness of the signal immediately upon generation with that observed further downstream, as experienced by the surface-attached cells, and observed virtually no decay in the fluorescent signal between the signal junction and the imaging region (Supplementary Fig. 3), only the time delay as calculated in Supplementary Table 1. While no fluid mixing occurs in this device—we operate under laminar flow regimes and there is no Lagrangian mixing—the diffusion of nutrients (or sodium fluorescein, which is 2–4 times the molecular weight of an amino acid) could potentially smooth out our nutrient signal. Diffusion can be further aided by Taylor dispersion, a phenomenon in which velocity gradients in the fluid flow (i.e., shear) work to increase the effective diffusion of a chemical species by spreading it across a larger region, thereby favoring diffusion. Were the magnitude of these effects non-negligible in our system, we should expect different slopes during transitions between the signals at the junction and downstream. Specifically, the downstream signal should have a longer transition time as we would detect fluorescence leaching into the $C_{low}$ phases of the signal. However, the time required to complete transitions (i.e., time to go from baseline to saturated fluorescent signal and vice versa) was about 2 s in both locations (Supplementary Fig. 3). Thus, our flow rates are sufficiently fast to carry our intended signal across the entire length of the device without noticeable smoothing from diffusion. We also determined that the periodic oscillations in the nutrient signal are robust across time. The programmed period ($T = 30$ s) was reliably quantified between peaks and between troughs from the repetitive fluorescein signal (Supplementary Fig. 3).

**Microfluidics experimental procedure**. The system used to operate the MSG involves: (1) a Nikon Eclipse Ti inverted microscope, (2) a full-case incubator that maintains a stable temperature (37 °C) around the entire microscope, except for the camera and light sources, (3) a computer to operate the microscope software (Nikon Elements) and MATLAB, (4) a data acquisition (DAQ) device that interfaces with MATLAB to control two pressure regulators, one for each nutrient source, (5) two

reservoirs of nutrient medium, one of each nutrient concentration, and (6) a source of compressed air. The compressed air is fed into the pressure system through a manual regulator, which caps the pressure directed toward the two automated regulators at 1.5 psi. To ensure that the automated regulators receive a stable input, the pressure of the compressed air source is higher than this maximum value. Each automated regulator is connected to and modulates the internal pressure of one reservoir of nutrient medium ($C_{high}$ or $C_{low}$). Each nutrient reservoir is a septum-capped glass vial (vials: VWR, cat. no. 548-0154; caps: VWR, cat. no. 548-0872) with two needles inserted into the silica septum: one short and one long. The short needle directly connects an automated pressure regulator with the air space within its reservoir, thereby adjusting the pressure within the reservoir as dictated by the MATLAB signal (Supplementary Fig. 1a). The long needle connects the fluid within its reservoir with the microchannel via tubing inserted into the inlets of the device (Supplementary Fig. 1a). The microscope and media are contained within a custom LIS incubator, which maintains the sample and all media at 37 °C.

Experiments were based on the exposure of cells attached to the lower surface of the microchannels to precisely controlled fluctuating or steady nutrient conditions, and the imaging of thousands of cells in the downstream imaging region in order to calculate their individual growth rates. The treated, dry microchannels were inoculated with ~50 µL of the seed culture (see Cell Preparation) for 10–15 min, allowing cells to settle and attach to the glass surface within each microchannel before the flow was established. Prior to inoculation, the microfluidic device was placed in a vacuum for at least 10 min to remove air from the PDMS. This step helped to avoid the presence of bubbles inside the channels, by removing air from the PDMS so that any air introduced in the setup would be absorbed by the PDMS. Inputs to fluctuating conditions were two septum-capped glass vials (one each for high and low nutrient) from which flow was driven by a custom-built air pressure system (Supplementary Fig. 1a). Inputs to steady conditions were 10 mL plastic syringes (Codan) from which flow was driven by a syringe pump (Harvard Apparatus). Outputs for all conditions led to liquid waste receptacles. To avoid changes in pressure throughout experiments, we ensured that the waste tubing was sufficiently short to never become submerged by the rising level of media waste.

**Image acquisition**. Individual cells from all microchannel environments were imaged with phase-contrast microscopy using a Nikon Eclipse Ti inverted microscope equipped with an Andor Zyla sCMOS camera (6.5 µm per pixel) at ×60 magnification (×40 objective with an additional 1.5×), for a final image resolution of 0.1083 µm per pixel. This magnification was high enough to detect changes in growth between each image, yet low enough to image hundreds of cells per field of view. Each position was repeatedly imaged every 117 s (1:57 min), a time step sufficiently short to allow the acquisition of multiple time points along a growth curve (i.e., 10 time points in a 20 min cell cycle), yet infrequent enough to image a total of 40–50 positions within each time step. Imaging 10–15 positions within each of four parallel conditions (i.e., fluctuating and steady $C_{low}$, $C_{ave}$, and $C_{high}$) required roughly 1.5 min. Generally, ten imaging positions per condition allowed us to track 500–1000 or more cells per nutrient condition. We confirmed that growth rates were independent of a cell's position along the 10-mm-long region imaged within the microchannel (Supplementary Fig. 4) and therefore that cells experienced identical nutrient time series, regardless of location within the microchannel. The uneven time step (1:57 min as opposed to 2:00 min) was chosen to avoid potential aliasing effects, by sampling at various points along the nutrient period instead of repeatedly at the same few. Light exposure was limited to 20 ms per image, with the shutter only open during image capture. Image acquisition was fully automated through Nikon Elements, supplemented with the Nikon Perfect Focus System to prevent loss of focus due to vertical shifts in the sample.

**Image processing**. In preparation for analysis, image sequences from microfluidic experiments were first passed through a particle tracking step and a quality control step. First, a custom MATLAB particle tracking pipeline was developed to (1) read image data directly from Nikon Elements image files, (2) identify particles based on pixel intensity, (3) fit an ellipse to each particle and measure particle parameters (e.g., length, width) and (4) track individual particles through time. Second, to exclude errors from our analysis—for example, particles arising from noise (i.e., non-uniformity in the background) or particles that include more than one cell—a quality control step trimmed our tracked dataset, using size criteria and noise filters to exclude errors. The parameter values used in both steps ensured that 93–97% (depending on nutrient condition) of tracks derive from isolated single cells. The final output of these two steps is a data matrix containing parameter data (e.g., cell length) over time for hundreds of individual cells growing in isolation. Reducing our analysis to cells without neighbors allowed us to assume no accumulation or depletion of medium components, and no physical interactions between cells. Cells in contact were excluded from the analysis to avoid the possibility of metabolic interactions and imprecision in the measurement of cell size. Specifics regarding the particle tracking and quality control steps are available in the scripts (see Code availability). Images in figure panels were adjusted for visualization using Fiji[49].

#### Quantification and statistical analyses

*Calculating instantaneous growth rate,* μ. One widely used method to calculate growth rate is to consider single-cell growth an exponential process[12,48] and solve for instantaneous growth rate, *μ*. This definition of growth rate is used throughout this study. From the length and width measured during particle tracking, the instantaneous volume of each individual cell was approximated as a cylinder with hemispherical caps[19]. The approximated volumes were then used to compute instantaneous single-cell growth rates in terms of volume doublings per hour. Using $V(t + \Delta t) = V(t) \times 2^{\mu \Delta t}$, we calculated $\mu$ between each pair of time steps, with $\Delta t = 117$ s (imaging frame rate). Specifically, we took the natural logarithm of each volume trajectory and calculated the slope between each point. Dividing the slope by the natural log of 2 changes the base of the exponential from $e$ to 2. Thus, $\mu$ represents the exponential rate at which volume doubles.

*Accounting for day-to-day variability in growth rate*. While steady-state growth rates were generally reproducible (Supplementary Fig. 5b and Supplementary Table 6), we found that growth rate measurements performed on the same day (i.e., same seed culture) were moderately correlated (Supplementary Table 7). This correlation indicates that slight differences between the seed culture (which was different for each experiment) contributed to the differences in growth rate measured from identical conditions between experiments. Thus, when comparing growth rate across conditions (e.g., $G_{fluc}$ and $G_{ave}$), we compared measurements performed on the same day before comparing between experimental replicates, calculating the fraction of $G_{ave}$ represented by the measured $G_{fluc}$ from that same experiment before calculating statistics (i.e., mean and standard deviation of $G_{fluc}/G_{ave}$) across experiments. We used this same approach when comparing $G_{fluc}$ to $G_J$, which was calculated from each experiment's $G_{low}$ and $G_{high}$, and $G_{fluc}$ to $G_{low}$. The alternative approach for this comparison would be to calculate the mean and standard deviation between experimental replicates before calculating fractions (e.g., $G_{fluc}/G_{ave}$) and combining the error. Numerically, this alternative approach yields very similar results.

**Reporting summary**. Further information on research design is available in the Nature Research Reporting Summary linked to this article.

## Data availability

Raw mass spectral data is deposited in massIVE and accessible with the accession code MSV000087096. The KEGG database used to annotate mass spectral data is available at https://www.genome.jp/kegg/pathway.html restricted to organism eco. Raw image data that support this study are available from the corresponding author upon request. Datasets of measurements from raw images are the Source data of this manuscript that are available at https://doi.org/10.5281/zenodo.4697572[50]. The Source data can be directly input into the openly available source code to produce reported figures and calculations (see Code availability).

## Code availability

Image processing, data analysis, and plotting scripts are available at https://github.com/jkimthu/growing-up[51]. Scripts to automate fluctuating signal generation are available at https://github.com/jkimthu/under-pressure[52].

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

## Acknowledgements

We are grateful to Michael Laub, Katharina Ribbeck, and members of the Stocker and Ackermann groups for discussions regarding this work. Russell Naisbit provided critical readings that improved this manuscript. The NCM3722 strain of *E. coli* was a generous gift from Suckjoon Jun. This work was funded by the Gordon and Betty Moore Foundation through Marine Microbiology Initiative Investigator Award GBMF3783 (to R.S.) and Grant GBMF3801 (to R.S.), by the Simons Foundation through Grants 542395 (to R.S.), 608247 (to S.P.), and 542379 (to M.A.) as part of the Principles of Microbial Ecosystems (PriME) Collaborative, and by the Swiss National Science Foundation under Grant 315230_176189 (to R.S.).

## Author contributions

J.N., V.F., M.A., and R.S. designed the study; J.N. and V.F. designed the microfluidic system, pressure control system/software, and particle tracking software; J.N. performed the microfluidic experiments and analyzed the growth data; S.P. performed MS measurements and analyzed the metabolomics data; U.S. provided an overview on metabolomics data generation and analysis; J.N., M.A., and R.S. wrote the paper; all authors reviewed the paper before submission.

## Competing interests

The authors declare no competing interests.
