## [Peer Review File · Nature Communications]

REVIEWER COMMENTS

Reviewer #1 (Remarks to the Author):

Using a microfluidics experimental setup, the authors exposed *E. coli* to a fluctuating complex (LB) nutrient environment. They found that the cells grew up to 50% slower under fluctuations compared to an equivalent average nutrient level. This growth rate decrease is greater than predicted by considerations based on Monod's curve and Jensen's inequality, suggesting that other physiological inefficiencies are at play than the simple observation that growth rate increases disproportionately less than substrate uptake. Meanwhile, single nutrient shift experiments predicted an even larger growth reduction than was observed, suggesting that the cells have adapted mechanisms to retain a higher growth rate under fluctuating environments.

This experiment presents an exciting hypothesis about the adaptations of microorganisms to real environments where nutrient availability likely fluctuates due to numerous sources of environmental variation. The large sustained growth rate detriment that was observed under fluctuating environments would present a selection pressure that would drive adaptive mechanisms to maintain homeostasis despite fluctuations. Thus, the whole scenario seems quite plausible and appealing. The timescales of these fluctuations are of course in question, but the authors span a wide range of timescales in their experiment (30s to 60min). This range in timescales spans both pre-transcriptional and transcriptional adaptation mechanisms (which are expected to take around 5 minutes). The authors observed growth rate changes for nutrient fluctuations with a period above 5min, indicating that the cells respond at least on a 2.5-minute timescale.

On the whole, the work is very impressive and the results are quite clear. The only limitation of the work is that authors do almost nothing to characterize the physiological nature of the fluctuation-induced cell state to uncover the mechanisms that enable better adaptation to fluctuations. However, that work likely would require an entirely separate study. I only have some minor specific comments below.

Minor comments

- The images were acquired every 2 minutes, so faster fluctuations than 2.5 minutes could not be ruled out. From this perspective, it seems strange that such a long image period was chosen – were there technical limitations that led to this choice?
- There seem to be two versions of Supplementary Table 2
- The noisiness in measured growth rates – are these inherent to the cellular process or to the measurements?

- The growth experiments seem well-validated. The primary question would seem to be the characterization of the distinct growth physiological state proposed by the authors during adaptation to fluctuating environments. The authors clearly show that the adaptation time decreases. However, they do not investigate into the nature of the adaptation that enables faster transitions in growth. The timescale of fluctuations was faster than the typical hours-long transition of a lag phase during exponential growth in a shake flask. However, the transition to the proposed distinct physiological state took several hours, which would suggest that gene expression changes are required. Can the authors speculate on the experimental accessibility of investigating the expression levels of cells in this particular physiological state? Would this be possible in their microfluidics set up, or could the same state be generated in a larger culture?

Reviewer #2 (Remarks to the Author):

Nguyen and coworkers addressed the very interesting topic of how a bacterial cell grows under fluctuating nutrient conditions. The authors grow single cells in steady state conditions under feast and famine conditions and applied, in addition, of/on nutrient pulses. They studied the ability of the cells to adapt to these conditions. The study was performed on a microfluidic level (I'm no expert in this technology) and provided exciting new data and a wealth of information, which will be of high interest to the community of microbiologists. The core finding was that fluctuating nutrients lower the growth rate. Only in comparison to a single disturbance of nutrient loss or shift, on/off fluctuation of nutrients enable cells to grow faster.

The approach is a very mechanistic one but opens unusual views on cell growth and stimulates thinking on bacterial cell growth, which is still a much discussed topic. In some places the paper is repetitive and also a bit over interpretive. The study would also benefit from including findings from bulk experiments where the influence of nutrient transitions under steady state conditions has been investigated manifold and on different levels. Also the topic of cell heterogeneity was not discussed and the influence of nutrient stress or nutrient loss on cell cycling duration might be worthwhile to discuss. Perhaps there might be valuable conclusions if the findings of this study based on volume of cells are compared to studies that focused on the duration of replication and cell division.

Specific remarks can be found below.

L88: How? Citation? Under steady state conditions there are always cells with individual properties, e.g. cell cycling, cell aging, cell division, extrinsic and intrinsic noise, etc... Cells that do not fit into the situation are washed out and are not further investigated....

This remark is somehow too general....

L93: Something like this is occurring during synchronization of yeast growth when the metabolism is on the edge between aerobic and respiro-fermentative metabolism. This competition between the two metabolism types cause the synchronized fluctuations in oxygen use and carbon dioxide release under chemostat steady state conditions.

This is at least one (exceptional) example that even under chemostat steady state conditions and even when synchronized, cells are able to respond with fluctuating metabolisms within short time frames.

L125: According to Supplementary Information, no standard conditions were used here. How did this influence the experiments?

L171: For me it looks like cell division when nutrients are available (Figure 2), which is something E. coli is known to do. Since after cell division there is nutrient depletion, cell just do not go on with cellular biomass growth to obtain the critical cell mass to start the cell cycle anew. For me it looks like a delay and not changing growth rates....

L175: If I understood correctly, the growth rate was determined by cell volume change (L157). What about cell numbers per time? Are, under fluctuating conditions, the same cell numbers are produced as e.g. under average steady-state conditions?

L214-219: This seems to be a very mechanistic view on cell growth...

L277: Is this stabilization of the growth rate dependent on the duration of the fluctuations?

L293: This behavior was already used in the so called transient state reactor set ups, which were used widely for testing bulk growth behavior of strains at successively increasing dilution rates. Can those facts be related to the observations here?

L331: This model seems to be very mechanistic. I believe that generation and replication times should be included in such a model.

L336-341: This is somehow not comprehensively described and not really shown in Figure 6. Are you comparing here the behavior of cells between a single nutrient shift and fluctuating shifts or of steady growth and fluctuating shifts? Fig 6 relates to single nutrient shifts and the last sentence to steady state growth.

L343: What is meant by qualitative and quantitative? Figure 6 shows real data only until 60 min and the other data are only predicted. It should be possible to generate lab data for an experiment running for 4 days to verify this assumption.

L349: Is there any knowledge available on bulk transient steady state cultivated reactor experiments?

L371: Is there any literature available for this statement? This should be dependent at least on time, frequency and seriousness of the impact.

L375: This statement seems to me over interpreted and maybe misleading. You compare here growth behavior of a single shift (i.e. disturbance) with situations where carbon sources are always

available although fluctuating. It is probably obvious that under latter conditions bacteria grow much better. This is not a novel finding.

L384: I'm not sure if this statement is generally true. Certainly such deep insight on the single cell level is highly valuable but on the bulk scale much work was done, especially from biotechnologists or ecologists.

L395: Unclear: what is meant here?

L417: Unclear: why is this mentioned here? The organism was grown on glucose or LB.

L436: media

L439-442: Unclear: The media contain multiple carbon and energy sources in unchanged proportions. The concentration of these sources might change substrate uptake rates in cells but this also influences how the substrates are being metabolized (e.g. see overflow metabolism...)

L468: In batch, nutrients can be taken up one after the other due to specific carbon type depletion. However, in continuous flow the substrate composition does not change....

Please explain.

L626: Could you provide a number in percentage?

L834: The lighter lines are parallels?

L839: In Figure 2a: cells not always seem to follow the nutrient signal, e.g. between 6 and 7 hours. This does not seem to mirror in 2b.

L859: Is this an artificial term created for the interpretation of the lower growth rates of cells in fluctuating environments? If a delay in growth is the reason for the behavior of those cells, this term might not contribute to provide explanations...

Supplementary Information

SFig2: Headline does not describe a-c

SFig5: Last sentence: Is there no possibility to create a standard situation for these types of experiments?

SFig6: a: See remark main paper line 439-442.

The data are probably obtained from bulk measurements. How can this be compared to the single cell measurements shown in all other datasets?

Second sentence: This assumption seems to be very artificial. Carbon source uptake is facilitated by many processes among them transport systems of different affinities, which react differently to the various concentrations in the medium. In this study, complex carbon sources were used, so I'm not convinced if this simplified assumption can be applied.

SFig9: No growth rate values given

Unclear: the cell cycle is expected to need 12 h? Or at least 6 hours until cell division? This seems unrealistic, because cell cycling does not depend that much on nutrient conditions... It is also known for E.coli that replication takes about 60 min and only stays undivided if nutrient conditions are bad.

What do you mean by stabilization time? If the nutrient loss occurs within replication time, the cells obtain reserves, which they can use. This is the concept of reaching the 'critical cell size' before proliferation can start. How does this model include the 'critical cell size' concept?

Reviewer #3 (Remarks to the Author):

The manuscript by Nguyen and team describes an experimentally elegant analysis of growth rate variation during nutrient shifts of different temporal range (seconds to minutes). The work is novel and appears to have been well-designed and well-executed. The data produced is nicely presented and straightforward.

I do have two issues with the manuscript that would need work prior to publication in a journal such as Nat Comm.

(1) The results, such as the ones presented on Figures 3,4,5 are clear, and the interpretation of the authors is likely correct. But the authors offer no experimental evidence to demonstrate which mechanism or mechanisms is/are triggered under rapid nutrient fluctuations "novel growth physiology" derived from differential transcription, or translation, or protein turnover, or transport? Without identifying an underlying mechanism, these results only descriptive. The authors need to experimentally identify and demonstrate which mechanism is behind these growth rate fluctuations and what really causes the "novel growth physiology".

(2)The other issue I have with this manuscript has to do with the expectation that growth physiology derived from single nutrient based steady-state models has to match with growth physiology under rapid nutrient shifts. The authors state "These results demonstrate that growth rate in rapid fluctuations is qualitatively and quantitatively distinct from steady-state growth dynamics." This result/conclusion is really not surprising as either their cells will be experiencing a completely different steady-state (if that has been reached) or still be under transient conditions, that is, adapting to the shifts. I think the authors could instead of presenting this fact as something completely unexpected and profound, present it as an obvious condition derived from shifting bacteria from one condition to another.

RESPONSE TO REVIEWER COMMENTS

The Reviewers' comments are copied below in black. Our responses are in blue.

Response to Reviewer #1

Using a microfluidics experimental setup, the authors exposed *E. coli* to a fluctuating complex (LB) nutrient environment. They found that the cells grew up to 50% slower under fluctuations compared to an equivalent average nutrient level. This growth rate decrease is greater than predicted by considerations based on Monod's curve and Jensen's inequality, suggesting that other physiological inefficiencies are at play than the simple observation that growth rate increases disproportionately less than substrate uptake. Meanwhile, single nutrient shift experiments predicted an even larger growth reduction than was observed, suggesting that the cells have adapted mechanisms to retain a higher growth rate under fluctuating environments.

This experiment presents an exciting hypothesis about the adaptations of microorganisms to real environments where nutrient availability likely fluctuates due to numerous sources of environmental variation. The large sustained growth rate detriment that was observed under fluctuating environments would present a selection pressure that would drive adaptive mechanisms to maintain homeostasis despite fluctuations. Thus, the whole scenario seems quite plausible and appealing. The timescales of these fluctuations are of course in question, but the authors span a wide range of timescales in their experiment (30s to 60min). This range in timescales spans both pre-transcriptional and transcriptional adaptation mechanisms (which are expected to take around 5 minutes). The authors observed growth rate changes for nutrient fluctuations with a period above 5min, indicating that the cells respond at least on a 2.5-minute timescale.

On the whole, the work is very impressive and the results are quite clear. The only limitation of the work is that authors do almost nothing to characterize the physiological nature of the fluctuation-induced cell state to uncover the mechanisms that enable better adaptation to fluctuations. However, that work likely would require an entirely separate study. I only have some minor specific comments below.

We thank the Reviewer for this positive assessment of our work. We have indeed embarked in the characterization of the physiology associated with the response to fluctuations, however as the Reviewer remarks, that is by itself a very large study, that we see separate from the present one in scope and time. We have addressed all the minor comments below — thank you!

Minor comments

- The images were acquired every 2 minutes, so faster fluctuations than 2.5 minutes could not be ruled out. From this perspective, it seems strange that such a long image period was chosen – were there technical limitations that led to this choice?

We thank the Reviewer for this thoughtful question. The timing between images was limited by the number of total positions imaged during an experiment. To attain a large sample size (i.e., number of bacteria imaged) for four parallel conditions (fluctuating, steady low, steady average, steady high) assayed simultaneously, our imaging procedure repeatedly looped through 10 positions within each of four microchannels, so that we acquired a total of 40 images before returning to the first position to begin the loop again. Imaging 40 positions required approximately 90 seconds, of which most of that time was spent on stage movement and automated focus adjustment (the latter an un-renounceable component to ensure good image quality). Our imaging time step of 2 min ensured the image acquisition system never fell behind. It is in principle possible to go faster, but this would come at the cost of either a much more complex setup (e.g., multiple microscopes in parallel) or of more noisy data (e.g., just 1 imaging position per condition)

To point readers to an explicit justification of our choice of imaging frequency:

- **Line 280:** we now reference the Image Acquisition section of the Methods just after acknowledging our 2 min time step as a limit in our temporal resolution.
- **Lines 1129-1130:** we edited the Image Acquisition section to help clarify why the time step used between images of the same imaging position was chosen, adding: “Imaging 10-15 positions within each of four parallel conditions (i.e., fluctuating and steady C_{low} , C_{ave} , and C_{high}) required roughly 1.5 min.”

- There seem to be two versions of Supplementary Table 2

Thank you for catching this error, which arose from deciding between which of the two formats to use to present very similar data, and then ultimately deciding to share both but forgetting to rename one of them. We have corrected this duplicated version by re-labeling these two tables **Supplementary Table 2** and **7**. We also adjusted the order in which these tables appear in the Supplementary Information, in effort to be more consistent with when they are referenced in the Main Text. Accordingly, **Supplementary Table 7** reports the steady-state growth rate, G , measured from all experimental replicates for each nutrient condition, whereas **Supplementary Table 2** summarizes the data in **Supplementary Table 7** to report the averaged G for each nutrient condition.

Edited instances of this data in the Main Text:

- **Lines 295-296:** refers to Supplementary Table 2
- **Line 1178:** refers to Supplementary Table 7

Edited instances of this data in the Supplementary Information:

- **Line 128:** refers to Supplementary Table 2
- **Line 529:** refers to Supplementary Table 7
- **Lines 525-535:** Legend and Data for Supplementary Table 2
- **Lines 589-599:** Legend and Data for Supplementary Table 7

- The noisiness in measured growth rates – are these inherent to the cellular process or to the measurements?

We thank the Reviewer for this question: this is indeed a point we did not address in our manuscript. Biological noise and measurement noise both contribute to the noise in the instantaneous growth rate, μ , in our data. Biologically-sourced noise in growth rate derives from stochastic fluctuations in the expression level of metabolic enzymes, which generate fluctuations in the rate of metabolic reactions and are a fundamental source of phenotypic heterogeneity between single cells (Kiviet et al. *Nature*, 2014). Kiviet and colleagues quantified biological noise in the instantaneous growth rate of individual *E. coli* cells as the standard deviation over the mean growth rate measured from instantaneous growth rates within a window of time (Kiviet et al. *Nature*, 2014). They found that noise in single-cell growth rate varied between 20% and 40% of the mean instantaneous growth rate of each cell, both within a cell cycle and between cell cycles from the same cell lineage (e.g., between mother and daughter cells). Kiviet et al. found that noise in single-cell growth rates decreases with increasing growth rate, with a noise intensity of 20% for a growth rate of approximately 0.9 h^{-1} .

We observe this trend also in our data: conditions with higher single-cell growth rates exhibit less noise. We quantified noise with the method used by Kiviet and colleagues, estimating noise intensities of 42%-138%. Our most comparable growth rate to those studied by Kiviet et al. was measured from our steady C_{low} condition, which had a steady-state growth rate of 1.07 h^{-1} , only slightly faster than Kiviet et al.'s 0.9 h^{-1} growth rate yet roughly 6-fold higher in noise intensity.

The additional noise in our data (compared to Kiviet et al.) derives from measurement noise. We calculated growth rate in the same manner (fitting size to an exponential of base 2), yet differed in two ways:

1. First, because single-cell width varied between our growth conditions, we calculated growth rate from volume instead of length. Volume is noisier than length because volume also contains width, a noisier parameter as width could be as small as 1 μm . We imaged at a resolution of $0.1083 \mu\text{m}$ per pixel, and thus the loss of only one pixel from imaging noise contributed approximately a 10% change in width. Noise in the width measurement was amplified in volume, because the square of the width enters into the calculation of volume.
2. Second, because we wanted to observe growth rate responding to shifts in nutrient concentration, we fit an exponential to two data points instead of several points, accepting some noise for a more temporally-resolved growth rate. Kiviet and colleagues fit an exponential to data points spanning one third of the cell cycle time, thus smoothing over 20 min of growth data — a timescale that would be too large for us to use to study minute-scale nutrient fluctuations.

While noise complicates our ability to observe strong responses from individual single-cell growth rate trajectories, we were able to observe the responses reported by collecting and averaging the instantaneous growth rates from hundreds of single cells. Averaged, the growth

rate signals reported in this study revealed key differences in single-cell growth rate responses to rapid nutrient fluctuations — an observation possible thanks to our efforts to balance temporal resolution and sample size.

We now address the issue of noise in the manuscript and thank the Reviewer for prompting us to do so.

Lines 260-270 in the Supplementary Information: we added the above analysis as a new plot (**Supplementary Figure 7b**). In the corresponding legend, we describe the quantification of noise within single-cell growth rate, and specifically include “Noise intensity in our data is higher than estimates of inherent biological noise (0.2–0.4) from stochastic fluctuations in metabolism and growth ([Kiviet et al. *Nature* 2014]). We attribute this increase to measurement noise, primarily derived from fluctuations in measured width, which is smaller and thus noisier than length, and enters squared in the calculation of volume.”

References cited in this response:

- Kiviet, D. J., Nghe, P., Walker, N., Boulineau, S., Sunderlikova, V., & Tans, S. J. Stochasticity of metabolism and growth at the single-cell level. *Nature* 514, 376-379 (2014).

- The growth experiments seem well-validated. The primary question would seem to be the characterization of the distinct growth physiological state proposed by the authors during adaptation to fluctuating environments. The authors clearly show that the adaptation time decreases. However, they do not investigate into the nature of the adaptation that enables faster transitions in growth. The timescale of fluctuations was faster than the typical hours-long transition of a lag phase during exponential growth in a shake flask. However, the transition to the proposed distinct physiological state took several hours, which would suggest that gene expression changes are required. Can the authors speculate on the experimental accessibility of investigating the expression levels of cells in this particular physiological state? Would this be possible in their microfluidics set up, or could the same state be generated in a larger culture?

We are in complete agreement with the Reviewer that insights into the molecular mechanisms of this growth physiology would be exciting and informative. We are indeed in the process of studying these, but that represents a very large, separate effort. Here we follow the Reviewer’s request and offer our preliminary views. Because many cellular processes are (1) intricately interconnected and (2) vary with growth state, we believe that -omics methods are the ideal next step towards characterizing the physiological nature of the fluctuation-adapted physiology and how it may differ from physiologies in steady environments. We have begun to develop a batch culture system for transcriptome and proteome measurements in nutrient fluctuations; however, the development of and experiments with this system will not be ready soon. This batch system is important for rigorous studies of gene expression, as the microfluidics set-up has two limitations:

1. First, the amount of protein we can extract from the microchannels is too low for proteomics. Each device accommodates in the order of 10,000 surface-attached cells. A sample for proteomics requires at least 100 μg of extracted protein (information from Dr. Mak Saito, WHOI, an expert on proteomics and current collaborator). In our fastest growing conditions (largest cell volume), the average cell volume at birth is $6 \mu\text{m}^3$, giving us $60,000 \mu\text{m}^3$ (or $6 \times 10^{-5} \text{ mL}$) of cell mass per microchannel. *E. coli* contain about 200 mg of protein per mL (Elowitz et al. *Journal of Bacteriology* 1999), giving us 0.012 mg (or 12 μg) of protein per microchannel. This value is an upper end, which further does not consider losses during extraction, yet an order of magnitude lower than the 100 μg needed.

Protein expression data is important to consider in studies with minute-scale nutrient fluctuations. Both transcription and translation produce mRNA and protein on minute-timescales. However, mRNA degradation can be much faster than protein dilution (Shamir et al. *Cell* 2016), implying that a transcriptional readout alone is not a sufficient indicator of gene expression.

2. Second, we unfortunately do not have the capacity to run several fluctuating microchannels in parallel, a necessity for -omics samples as sampling is destructive. This is due to the fact that each fluctuating condition requires a relatively complex pressure-control system, which would require large effort and cost to replicate. This limitation prevents comparison from parallel microchannels, preventing for example gene expression measurements from successive periods of nutrient fluctuation (as measured microscopically in **Fig. 5a**). This limits our ability to determine changes in gene expression across the transition from a steady-state adapted physiology to a fluctuation-adapted physiology, as well as our ability to determine how stable the gene expression underlying the fluctuation-adapted physiology may be.

Overall, we fully agree with the Reviewer that understanding cellular expression levels is an exciting and important next step! While not feasible from our microchannel experiments, we are developing a batch system to rigorously assess RNA and protein expression in parallel steady and fluctuating environments.

References cited in this response:

- Elowitz, M. B., Surette, M. G., Wolf, P. E., Stock, J. B., & Leibler, S. Protein Mobility in the Cytoplasm of *Escherichia coli*. *Journal of Bacteriology* 181(1), 197-203 (1999).
- Shamir, M., Bar-On, Y., Phillips, R., & Milo, R. SnapShot: timescales in cell biology. *Cell* 164(6), 1302-1302 (2016).

All in all, we thank the Reviewer immensely for their feedback and especially for highlighting concepts (e.g., noise) that were not thoroughly discussed in the previous version of our manuscript. Their review enabled us to clarify and enrich our manuscript.

Response to Reviewer #2

Nguyen and coworkers addressed the very interesting topic of how a bacterial cell grows under fluctuating nutrient conditions. The authors grow single cells in steady state conditions under feast and famine conditions and applied, in addition, of/on nutrient pulses. They studied the ability of the cells to adapt to these conditions. The study was performed on a microfluidic level (I'm no expert in this technology) and provided exciting new data and a wealth of information, which will be of high interest to the community of microbiologists. The core finding was that fluctuating nutrients lower the growth rate. Only in comparison to a single disturbance of nutrient loss or shift, on/off fluctuation of nutrients enable cells to grow faster.

The approach is a very mechanistic one but opens unusual views on cell growth and stimulates thinking on bacterial cell growth, which is still a much discussed topic. In some places the paper is repetitive and also a bit over interpretive. The study would also benefit from including findings from bulk experiments where the influence of nutrient transitions under steady state conditions has been investigated manifold and on different levels. Also the topic of cell heterogeneity was not discussed and the influence of nutrient stress or nutrient loss on cell cycling duration might be worthwhile to discuss. Perhaps there might be valuable conclusions if the findings of this study based on volume of cells are compared to studies that focused on the duration of replication and cell division. Specific remarks can be found below.

We thank the Reviewer for their thoughtful and comprehensive review of our manuscript. Their comments and questions helped us to greatly improve the clarity and depth of our work. In our specific responses below (please see specific responses for line numbers), we made revisions to:

- Streamline our writing to avoid repetition.
- Edit out areas that were too interpretive, focusing instead on the specific contributions of our work.
- Draw stronger connections between our findings and those from bulk experiments, to demonstrate how our work connects with the rich body of work that already exists.
- Provide data and discussion on cell cycle duration (which we often refer to as “division time” here), to enable direct connections between volumetric growth and population growth.

L88: How? Citation? Under steady state conditions there are always cells with individual properties, e.g. cell cycling, cell aging, cell division, extrinsic and intrinsic noise, etc... Cells that do not fit into the situation are washed out and are not further investigated....This remark is somehow too general....

We thank the Reviewer for pointing our attention to a sentence that was indeed rather general and poorly informative. We have edited this sentence to be more specific about how the gene expression of *E. coli* is dependent on nutrient availability (including a reference), as can be seen now on **Lines 108-110**, which read:

“Growing *E. coli* physiologically transition between steady states of growth by adjusting their expression of anabolic and catabolic pathways based on nutrient availability ([Erickson et al., *Nature* 2017]).”.

L93: Something like this is occurring during synchronization of yeast growth when the metabolism is on the edge between aerobic and respiro-fermentative metabolism. This competition between the two metabolism types causes the synchronized fluctuations in oxygen use and carbon dioxide release under chemostat steady state conditions.

This is at least one (exceptional) example that even under chemostat steady state conditions and even when synchronized, cells are able to respond with fluctuating metabolisms within short time frames.

Thank you for bringing up this particularly relevant example! We have revised our manuscript to fold it into our discussion on **Lines 778-783**, along with a few relevant citations.

L125: According to Supplementary Information, no standard conditions were used here. How did this influence the experiments?

We thank the Reviewer for this important question. In this study, we vary the *concentration* of LB medium to vary growth rate, whereas -- as the Reviewer points out -- it is more standard in the literature to vary the nutrient *composition* (e.g., glucose, glucose + amino acids, tryptic soy broth (TSB), etc.) (Schaechter et al. 1958; Scott et al. 2010; Taheri-Araghi et al. 2015). We did this because our goal was explicitly to study how rapid changes in nutrient *concentration* affect growth.

We have now revised our manuscript to include an explicit validation that the conditions we tested produce growth behaviors consistent with those previously established from work with *E. coli* in diverse media. The revision includes:

- Two new Supplementary panels (**Supplementary Fig. 5 c,d**) that specifically compare our data to that reported by Taheri-Araghi et al. 2015, in which similar single-cell measurements (based on microfluidics and time-lapse microscopy) are performed in steady environments across a range of defined and undefined nutrient media. We show thereby that our conditions (i.e., the use of different LB concentrations) replicate the relationships between (1) growth rate and cell size and (2) growth rate and division time reported by Taheri-Araghi and colleagues.
- An explicit statement in the Main Text on **Lines 212-216**: “We found that varying nutrient concentration reproduced key relationships between growth rate, cell size and division time (**Supplementary Fig. 5c, d**) previously established by varying the nutrient source (e.g., glucose vs. tryptic soy broth) (Taheri-Araghi et al. 2015), suggesting that variations in growth rate due to changes in nutrient concentration and nutrient source are physiologically similar.”

In summary, our data on bacterial growth across various concentrations of LB replicates the relationship between all cell cycle parameters (growth rate, division time, volume at birth) previously seen by varying nutrient media. We believe that the consistency in these relationships between our conditions and others demonstrates that our work is relevant to the framework established by prior fundamental works in bacterial growth (Schaechter et al. 1958; Koch 1971; Scott et al. 2010; Taheri-Araghi et al. 2015).

References cited in this response:

- Schaechter, M., Maaløe, O., & Kjeldgaard, N. O. Dependency on medium and temperature of cell size and chemical composition during balanced growth of *Salmonella typhimurium*. *Microbiology* 19(3), 592-606 (1958).
- Scott, M., Gunderson, C. W., Mateescu, E. M., Zhang, Z., & Hwa, T. Interdependence of cell growth and gene expression: origins and consequences. *Science* 330(6007), 1099-1102 (2010).
- Taheri-Araghi, S., Bradde, S., Sauls, J. T., Hill, N. S., Levin, P. A., Paulsson, J., ... & Jun, S. Cell-size control and homeostasis in bacteria. *Current Biology* 25(3), 385-391 (2015).
- Koch, A. L. The adaptive responses of *Escherichia coli* to a feast and famine existence. *Advances in Microbial Physiology*. 6, 147-217 (1971).

L171: For me it looks like cell division when nutrients are available (Figure 2), which is something *E. coli* is known to do. Since after cell division there is nutrient depletion, cell just do not go on with cellular biomass growth to obtain the critical cell mass to start the cell cycle anew. For me it looks like a delay and not changing growth rates....

We thank the Reviewer for raising this alternative hypothesis, which we did not address or rule out in the previous version of our manuscript. If our interpretation of their comment is correct, the Reviewer's hypothesis implies that our observed growth rate fluctuations may arise not from nutrient-induced changes in single-cell volumetric growth rate (as we conclude), but rather from changes in the frequency of cell division. Specifically, if cells in the minutes immediately following a division cease to accumulate biomass and if cells generally divide in high nutrient conditions, a higher frequency of cell division in C_{high} may lead to more cells with a growth "delay" in C_{low} and the observation of a lower average growth rate in C_{low} . Under this hypothesis, cells that are growing in fluctuating environments add volume at a single volumetric growth rate (regardless of external nutrient concentration), yet the averaged volumetric growth rate in C_{low} is lower because a larger fraction of cells may be post-divisional and have a volumetric growth rate of near-zero (due to the delay mentioned by the Reviewer).

To test this hypothesis, we performed a new analysis that shows that we can rule this hypothesis out, and include it in our revised manuscript as **Supplementary Fig. 8**. We reasoned: if volumetric growth rate μ does not change in direct response to nutrient concentration, but rather is indirectly affected by how nutrient concentration affects cell division, we would expect volumetric growth rate to be determined by whether cells are pre- or post- cell division. Thus, we identified all cell division events from *E. coli* growing under 60 min nutrient

fluctuations. For cells that began and completed cell division in the same nutrient concentration, we analyzed volumetric growth rates immediately before and after cell division, allowing us to disentangle the contributions of cell division and nutrient concentration on our observed fluctuations in average volumetric growth rate (**Fig. 2b, c**).

The Reviewer is indeed correct that cell division mostly occurs in C_{high} and that cell division appears to incur a lag time when cells experience C_{low} (**Supplementary Fig. 8a**); however, we did not observe a post-divisional delay in volumetric growth rate (**Supplementary Fig. 8b**). Instead, volumetric growth rate in 60 min nutrient fluctuations was relatively stable before and after cell division, confirming that volumetric growth rate was dependent on the immediate nutrient concentration rather than on the stage of the cell cycle (**Supplementary Fig. 8b**).

In **Supplementary Fig. 8b**, we measured the instantaneous growth rate (μ , the rate at which single cells double in volume) of single cells in the minutes before and after cell division. We considered cells that divided in one nutrient condition (e.g., C_{high}) and remained in that same condition for at least 4 min after cell division. For cells dividing in C_{high} , the average growth rate in the 2 and 4 min after cell division was $2.10 \pm 0.05 \text{ h}^{-1}$ and $2.06 \pm 0.06 \text{ h}^{-1}$, respectively. While lower than the average growth rate in the 2 and 4 min prior to cell division ($2.43 \pm 0.05 \text{ h}^{-1}$ and $2.61 \pm 0.05 \text{ h}^{-1}$, respectively), the post-division growth rates were still substantially greater than zero. The same was true for cells dividing in C_{low} , where the average growth rate 2 min post-division was $0.72 \pm 0.17 \text{ h}^{-1}$, which was not different from the growth rate 2 min prior to division ($0.85 \pm 0.17 \text{ h}^{-1}$) (**Supplementary Fig. 8b**).

This result deviates from what would be expected from the Reviewer's hypothesis: we did not observe cell division-dependent growth delays that would explain our observed fluctuations in average volumetric growth rate (**Fig. 2b,c**). Instead, volumetric growth rates both pre- and post-division were consistent with the averaged growth rate values we measured from fluctuation-adapted cells experiencing either C_{high} or C_{low} (**Fig. 4d**).

Overall, the observation that volumetric growth rate depends on nutrient concentration and not on the timing relative to cell division is consistent with the conclusion in our manuscript that minute-scale nutrient fluctuations induce fluctuations in the rate at which single cells accumulate cell volume. We are grateful to the Reviewer for bringing up this alternative hypothesis for the observed growth rate under fluctuations, which we think helped us to more thoroughly support our conclusions.

Lines 254-257: We have added a sentence to our revised manuscript, referring to this new Supplementary Figure, to rule out this alternative hypothesis:

“While cell divisions occurred more frequently during phases of high nutrient, we confirmed that the fluctuations in growth rate reflected responses in single-cell volume, rather than cell division responses (**Supplementary Fig. 8**)”.

L175: If I understood correctly, the growth rate was determined by cell volume change (L157). What about cell numbers per time? Are, under fluctuating conditions, the same cell numbers produced as e.g. under average steady-state conditions?

We thank the Reviewer for their broad and thorough consideration of bacterial growth. The Reviewer is correct in understanding that our manuscript focuses on cell volume change, the definition of “growth rate” used throughout this study. And as the Reviewer points out, another key definition of bacterial growth rate measures the growth of population size by quantifying the rate at which cell division produces new cells.

Lines 237-239: We have added to our revised manuscript the following explanation to explicitly state why our choice to focus on volumetric growth rate was specifically made to study second and minute scale fluctuations:

“We focused on single-cell volumetric growth rather than cell replication or division times, because biomass production has been shown to respond within minutes of a nutrient shift whereas cell division responds more slowly (within an hour)” (Kjeldgaard et al. 1958).

We very much appreciate the Reviewer’s interest in cell division, though, and have revised our manuscript to include data that answers their specific questions, in particular:

- **Cell numbers per time:** we have added **Supplementary Fig. 8d** to exemplify the number of individual cells tracked over the course of a single experiment. We would like to note that, unlike in batch systems, cell number in our microfluidic experiments depends on cell attachment and detachment, the dynamics of which vary with growth condition.
- **Cell production in fluctuating vs steady average conditions:** we have added **Supplementary Fig. 8c** to directly show the production rate of new cells in fluctuating and steady conditions. Because different cell numbers are tracked in different conditions, we compare the number of cell divisions recorded per 1000 cells as a function of time.

All in all, nutrient fluctuations produce responses in both volumetric growth rate (biomass added) and population growth rate (“birth” of new cells by cell division) (**Fig. 2; Supplementary Fig. 8**). Both are valuable parameters of bacterial growth with broad implications. While our manuscript focuses on volumetric growth, we thank the Reviewer for the attention to population growth rate and the opportunity to address it. We feel that these revisions to include cell division data may help readers make direct connections between our primary findings with volumetric growth to systems that primarily measure cell number or optical density. We are additionally working on a separate manuscript that focuses specifically on the effect of nutrient fluctuation timescale on single-cell size.

Lines 254-257: we have revised these lines to point readers to the analyses of cell division mentioned above. They now read:

“While cell divisions occurred more frequently during phases of high nutrient, we confirmed that the fluctuations in growth rate reflected responses in single-cell volume, rather than cell division responses (**Supplementary Fig. 8**).”

L214-219: This seems to be a very mechanistic view on cell growth...

We thank the Reviewer for highlighting this point of poor clarity in our manuscript. We did not intend to suggest that this model was expected or likely to fully describe bacterial growth. Indeed, such would be a considerable oversimplification of a complex cellular process. We had instead intended simply to use this mathematical model in our discussion to help readers begin to intuit the physiological meaning of a reduction in growth. Specifically, why does it make sense that temporal fluctuations at minute timescales are not averaged by cells?

In effort to avoid confusing or misleading our readers with this model, we revised our language in this discussion. We sought to be more specific in our use of this model as a conceptual guide, rather than as an established framework in the field from which we are trying to claim novelty. For example:

- **Lines 327-328:** we revised our prior introduction of the model as a potential “explanation” to instead frame it as a “mathematical model” that “highlights the physiological implications of fluctuating nutrients”.
- **Lines 328-334:** we revised the language describing predictions of this model to avoid sounding as if we were highlighting an unexpected result. Rather it now reads simply as a description of the model and what it predicts.
- **Lines 336-340:** we similarly replaced the language comparing the model with our results to make it clear that deviation from this model is not a result we are claiming as novel. This model is simply a means of quantitatively conceptualizing what a specific growth rate value might mean.
- **Line 398-399:** we edited “the reduction in G_{fluc} from G_J ” to only state “the reduction in G_{fluc} ” to avoid suggesting that G_J was the expected result.

We thank the Reviewer again for highlighting these lines as a cause of confusion, allowing us to revise and improve our manuscript. We hope that our revisions serve to make our scientific contribution more clear: that minute-scale nutrient fluctuations are detrimental to growth rate, but not as detrimental as one might think due to a fluctuation-adapted physiology that alleviates a substantial portion of the potential loss.

L277: Is this stabilization of the growth rate dependent on the duration of the fluctuations?

We thank the Reviewer for highlighting points to help us improve our communication and clarity in this section of our manuscript.

- **Lines 411-412:** We have edited this section to explicitly state in the main text the stabilization time observed from cells grown under $T = 15$ min and $T = 60$ min

fluctuations: “We observed this stabilization of growth rate in fluctuating environments on 15 and 60 min periods...”

- **Lines 410-485:** We also substantially revised our discussion of these results to better elaborate our interpretation of growth rate dynamics under fluctuating nutrient conditions. This section was rather redundant in certain areas and we reorganized it in effort to make it more clear, direct and concise.
 - Within this, **Lines 475-479** have been added to specifically address the potential differences in growth rate dynamics (and therefore growth physiology) associated with nutrient fluctuations of different timescales:

“That cells grown in $T = 15$ min and 60 min nutrient fluctuations induced growth rate responses, distinct from cells responding to single shifts, suggests the existence of a distinct physiology that cells adopt upon experiencing rapid nutrient fluctuations. Growth rate stabilized at comparable values in both fluctuating timescales (Fig. 4d), suggesting that $T = 15$ min and 60 min nutrient fluctuations induce the same or very similar physiologies.”

In short, our data indicate that both fluctuating timescales induced a similar growth rate response, including similar timescales of growth rate stabilization. Future studies (i.e., bulk - omics measurements) on growth responses to nutrient fluctuations will determine whether $T = 15$ min and $T = 60$ min fluctuations induce an identical fluctuation-adapted growth physiology, or whether different fluctuations produce a spectrum of physiologies.

L293: This behavior was already used in the so called transient state reactor set ups, which were used widely for testing bulk growth behavior of strains at successively increasing dilution rates. Can those facts be related to the observations here?

We thank the Reviewer for drawing our attention to the parallels between our work and that of reactor set-ups, particularly the example of metabolic fluctuations in yeasts.

Lines 776-783: We have now included a discussion related to our single-cell growth observations with the understanding of metabolic fluxes in transient state conditions and feel that this addition serves to enrich our manuscript. Thank you!

L331: This model seems to be very mechanistic. I believe that generation and replication times should be included in such a model.

We thank the Reviewer for their careful attention to the multiple parameters by which bacterial growth rate is measured. As noted, our study uses “growth rate” to mean the rate at which a single cell doubles in volume, often considered a fundamental parameter to the physiology of single cells (Taheri-Araghi et al., 2015; Harris & Theriot, 2019; Wallden et al., 2019) and the primary focus of this study. We appreciate, though, that this parameter is specific to single-cell studies and that information on other parameters, such as cell replication times, can facilitate stronger connections between the findings from our work and related work in other systems.

To facilitate these connections, we have now edited our manuscript to include a full description of cell replication times (called “division time” in our manuscript), including:

Lines 278-314 in the Supplementary Information: we have added a new figure (**Supplementary Fig. 8**) quantifying cell division in steady and fluctuating environments.

- As described in earlier responses, **Supplementary Fig. 8a-d** demonstrates that while probability that a cell will divide is higher in C_{high} than C_{low} , changing the frequency of cell division (that is, changing the production rate of new cells) does not contribute to changes in single-cell growth rate (rate of volumetric growth).
- **Supplementary Fig. 8e** visualizes the distribution of division time measured from each steady and fluctuating condition. The figure legend details the mean, standard deviation, and number of unique division events measured per condition.

These data support a statement we added to the main text on **Lines 255-257**, stating that “the fluctuations in growth rate reflected responses in single-cell volume, rather than cell division responses (**Supplementary Fig. 8**)”.

Lines 236-239: we have also added a sentence in the main text to explicitly justify our primary focus on volumetric growth: “We focused on single-cell volumetric growth rather than cell replication or division times, because biomass production has been shown to respond within minutes of a nutrient shift whereas cell count responds more slowly (within an hour) [Kjeldgaard et al., 1958]”. Because volume and cell division respond to nutrient shifts on different timescales, we feel that considering both in the same model is a whole other undertaking in itself.

We hope that our revision to include a full description of division time fulfills two purposes:

- (1) Instilling greater confidence regarding our conclusions and why we chose to focus on single-cell volumetric growth.
- (2) Enhancing the ease with which readers working in bulk systems can draw direct connections between our findings and their systems.

Overall, we feel that these revisions have very much improved our manuscript and are thankful that the Reviewer brought this to our attention.

References cited in this response:

- Taheri-Araghi, S., Bradde, S., Sauls, J. T., Hill, N. S., Levin, P. A., Paulsson, J., ... & Jun, S. Cell-size control and homeostasis in bacteria. *Current Biology* 25(3), 385-391 (2015).
- Harris, L. K., & Theriot, J. A. Relative rates of surface and volume synthesis set bacterial cell size. *Cell* 165(6), 1479-1492 (2016).
- Wallden, M., Fange, D., Lundius, E. G., Baltekin, Ö., & Elf, J. The synchronization of replication and division cycles in individual *E. coli* cells. *Cell* 166(3), 729-739 (2016).

L336-341: This is somehow not comprehensively described and not really shown in Figure 6. Are you comparing here the behavior of cells between a single nutrient shift and fluctuating shifts or of steady growth and fluctuating shifts? Fig 6 relates to single nutrient shifts and the last sentence to steady state growth.

We thank the Reviewer for noting that this portion of our manuscript was unclear. Here, we are comparing growth rate responses to fluctuations and single shifts. Our intention was to describe the null model producing the analysis in **Fig. 6** in the prior paragraph (now **Lines 547-558**) and the results in the current (now **Lines 560-770**). However, we can see now that our previous version was rather dense with technical details, which obscured clear communication. We have revised both paragraphs in effort to make this more clear through the following revisions:

- **Lines 548-551**: we edited our description of the null model, which is built from single shift behavior (grey points in Fig. 6), to be more concise.
- **Lines 551-558**: we moved most of the technical details of the model to a new section in the Supplementary Information, titled “Supplementary Procedures: Construction of Null Model”. This section immediately follows **Supplementary Fig. 10**, which it describes, making this new placement better able to facilitate a careful understanding of the null model. In the main text, we replaced these details with a clearer summary of the null model, which we hope provides a better transition into **Fig. 6** and our discussion of its results.
- **Fig. 6a**: we added another panel to **Fig. 6** to include a visual description of the models used to compare measured and predicted growth rate in fluctuations in the original **Fig. 6** (now **Fig. 6b**). We hope that the visual, which parallels the visualization of single shift growth rate responses in **Fig. 4a**, helps readers see how the single shift data were used to produce the predicted growth rates in **Fig. 6b**.

We thank the Reviewer again for pointing out a lack of clarity and hope that these edits allow readers to easily understand how these results were obtained.

L343: What is meant by qualitative and quantitative? Figure 6 shows real data only until 60 min and the other data are only predicted. It should be possible to generate lab data for an experiment running for 4 days to verify this assumption.

The Reviewer makes an excellent suggestion, and we agree: a longer experiment would be ideal to add real data to validate our predictions. Unfortunately, our microfluidic method cannot run experiments for much longer than 10 hours due to cell loss from the device over time. Our measurements rely on cell attachment to the surface of our microchannels. Because cells detach from the device, we often do not have many individuals remaining much longer than 10 hours, particularly in the fluctuating channels (**Supplementary Fig. 8d**).

Lines 763-766: We elaborated on “qualitatively and quantitatively” to specify the difference we observed between predicted and measured growth rates across fluctuating timescales and to

directly connect these results with the conclusions highlighted in our abstract. The lines now read:

“These results demonstrate that growth rate in rapid fluctuations is not only quantitatively distinct from the values of G_{fluc} predicted from single-shift dynamics, we measured a qualitatively different trend in G_{fluc} across nutrient timescales.”

L349: Is there any knowledge available on bulk transient steady state cultivated reactor experiments?

Indeed there is! We thank the Reviewer for encouraging us to draw stronger links between our single-cell work and prior studies from batch or continuous culture conditions.

Lines 776-786: We have added a short discussion on how the particular example of aerobic and respiro-fermentative growth, mentioned by the Reviewer in an earlier comment (L93), highlights mechanisms by which external nutrient fluctuations produce changes in microbial growth rate. We connect this example with bulk work we have performed in *E. coli* in similar pulsing glucose conditions (Sekar et al., 2020).

References cited in this response:

- Sekar, K., Linker, S. M., Nguyen, J., Grünhagen, A., Stocker, R., & Sauer, U. (2020). Bacterial glycogen provides short-term benefits in changing environments. *Applied and Environmental Microbiology*, 86(9).

L371: Is there any literature available for this statement? This should be dependent at least on time, frequency and seriousness of the impact.

Line 831: We've added references to the statement in mention, specifically Erickson et al., 2017; Kjeldgaard et al., 1958; Mori et al., 2017; Kohanim et al., 2017. We have also amended the statement to better accommodate the important dependencies mentioned by the Reviewer.

References cited:

- Erickson, D. W., Schink, S. J., Patsalo, V., Williamson, J. R., Gerland, U., & Hwa, T. A global resource allocation strategy governs growth transition kinetics of *Escherichia coli*. *Nature* 551(7678), 119 (2017).
- Kjeldgaard, N. O., Maaløe, O., & Schaechter, M. The transition between different physiological states during balanced growth of *Salmonella typhimurium*. *Microbiology* 19(3), 607-616 (1958).
- Mori, M., Schink, S., Erickson, D. W., Gerland, U., & Hwa, T. Quantifying the benefit of a proteome reserve in fluctuating environments. *Nature Communications*, 8(1), 1225 (2017).
- Kohanim, Y. K., Levi, D., Jona, G., Towbin, B. D., Bren, A., & Alon, U. A bacterial growth law out of steady state. *Cell Reports* 23(10), 2891-2900 (2018).

L375: This statement seems to me over interpreted and maybe misleading. You compare here growth behavior of a single shift (i.e. disturbance) with situations where carbon sources are always available although fluctuating. It is probably obvious that under latter conditions bacteria grow much better. This is not a novel finding.

We thank the Reviewer for highlighting this statement as a potential point of miscommunication.

As written in our original manuscript, it read: “Here, we report that rapidly fluctuating nutrient environments induce a novel bacterial physiology with growth responses fundamentally distinct [from] single shifts from steady-state that enable bacteria to grow faster in fluctuations than expected from the existing paradigm.”

We have revised our text to be more specific about the “existing paradigm”, by which we meant that single shifts are currently used as experimental models to study bacterial growth in fluctuating environments. Given the ongoing application of single shifts to study fluctuations (Mori et al. *Nature Communications*, 2017; Kohanim et al. *Cell Reports*, 2018; Basan et al. *Nature*, 2020), we argue that it is not in fact “obvious” that bacteria should grow differently under fluctuating conditions.

Given other known bacterial adaptations to environmental change, such as chemotaxis and condition-dependent gene expression, we agree that it is reasonable that bacteria may have evolved adaptations for growth in rapid nutrient fluctuations. Yet, these adaptations have not been observed before: our study is the first to systematically search for and describe these adaptations, a step made possible by the novel technology we have developed. We find that growth responses to single shifts occur on different timescales and reach different growth rates than growth responses to the same shifts applied repeatedly over time, i.e. fluctuations. We propose that the different growth response observed from fluctuating environments is an evolved adaptation because it enables bacteria to grow faster in fluctuating environments than they would without. We do not believe that this is an obvious finding.

This statement in our conclusion now reads:

Lines 831-837: “Despite the evidence that bacteria encounter rapid and repeated nutrient fluctuations in their environments (1; 2; 4; 5; 8; 37; 38; 39; 40; 41), single shifts in nutrient composition remain the dominant method by which bacterial growth is studied in dynamic nutrient conditions (11; 24; 25). This study reports key differences in growth rate dynamics between fluctuations and single shifts, thus introducing repeated fluctuation as a better experimental system to study variability in microbial habitats than the classic single-shift paradigm.”

References cited in this review:

- Mori, M., Schink, S., Erickson, D. W., Gerland, U., & Hwa, T. Quantifying the benefit of a proteome reserve in fluctuating environments. *Nature Communications*, 8(1), 1225 (2017).

- Kohanim, Y. K., Levi, D., Jona, G., Towbin, B. D., Bren, A., & Alon, U. A bacterial growth law out of steady state. *Cell Reports* 23(10), 2891-2900 (2018).
- Basan, M., Honda, T., Christodoulou, D., Hörl, M., Chang, Y.F., Leoncini, E., ... Sauer, U. A universal trade-off between growth and lag in fluctuating environments. *Nature* 584(7821), 470-4 (2020).

L384: I'm not sure if this statement is generally true. Certainly such deep insight on the single cell level is highly valuable but on the bulk scale much work was done, especially from biotechnologists or ecologists.

The Reviewer makes an excellent point that studies performed at bulk scales have long observed bacterial physiology in a wide array of nutrient environments. In the previous version of our manuscript, we had written:

“Only through direct experiments designed to test the role of nutrient fluctuations in a controlled manner could nutrient timescale emerge as a key parameter for bacterial growth.”

In this statement, we did not intend to suggest that work from bulk studies could not contribute value. We rather wished to emphasize that experimentally performing repeated fluctuations enabled the observation of growth responses that differed from otherwise identical (but not repeated) single shifts in nutrient.

Lines 839-841: We have replaced this sentence with another that more specifically states our intended meaning:

“In finding starkly different responses between fluctuations and single shifts, our study presents a conceptual advance important for the interpretation of past work and the direction of future work considering microbial growth in complex environments.”

L395: Unclear: what is meant here?

We thank the Reviewer for pointing out various parts of our submitted manuscript that were unclear, including this one. We tried to address each part carefully, as the accessibility of our language and ideas is important to us.

In this particular instance, we had intended to say that our study varied only the nutrient timescale but that many other patterns of fluctuations exist in nature. This idea is also encompassed in the next sentence so we opted to remove this more nuanced sentence, which caused some confusion (deletion is tracked on **Line 894**).

L417: Unclear: why is this mentioned here? The organism was grown on glucose or LB.

We had intended this statement to say that our chosen *E. coli* strain is recommended for physiological studies in various media, as it lacks the growth defects characterized in other common lab strains of *E. coli*, such as MG1655.

Lines 914-915: We have edited this section to say this more explicitly:

“[The strain] ... is derived from the background strain, K-12 NCM3722, which lacks the growth defects observed in other strains of *E. coli* (44; 45).”

L436: media

Line 950: Thank you! We have corrected “mediums” to “media”.

L439-442: Unclear: The media contain multiple carbon and energy sources in unchanged proportions. The concentration of these sources might change substrate uptake rates in cells but this also influences how the substrates are being metabolized (e.g. see overflow metabolism...)

We agree with the Reviewer that this statement was unclear and misleading, suggesting that there were no changes in the way the nutrients were being metabolized across nutrient conditions.

Lines 953-956: To more accurately describe our results, we modified the manuscript to state simply that we do not see changes in the preference of which metabolites are consumed:

“we confirmed with metabolomic profiling that the different steady-state growth rates between the three steady conditions resulted from concentration-dependent changes in nutrient uptake rates, rather than changes in preferential metabolite uptake (**Supplementary Fig. 6**).”

L468: In batch, nutrients can be taken up one after the other due to specific carbon type depletion. However, in continuous flow the substrate composition does not change.... Please explain.

The Reviewer makes an important point, which is that the conditions in our microfluidic setup are different than those in batch cultures and that the single-cell observations involve a constant supply of nutrients, which continuously replenishes all resources available. As the Reviewer also points out, nutrients in batch culture can be consumed in a preferential order and the concentrations of different components of the medium may affect which nutrients are taken up preferentially (as the Reviewer notes in their comment regarding **Supplementary Fig. 6**). This potential issue is what had motivated us to determine whether the various nutrient conditions used in our study differed in which carbon sources were consumed preferentially, by using metabolomics from batch cultures to monitor metabolite depletion from the three conditions (C_{low} , C_{ave} and C_{high}) used in our microfluidics experiments. This work is presented in **Supplementary Fig. 6** of the original manuscript).

In testing whether the preferences in carbon sources and the rate of consumption between nutrient conditions is consistent at early stages in batch culture, we saw no evidence that the preferred carbon sources varied between the different nutrient conditions. We elaborate on our interpretation of this data in our response to the comment on **Supplementary Fig. 6** (below).

Lines 962-972: we edited the manuscript to improve our explanation of the metabolomic experiments and results:

“Changes in extracellular nutrient concentrations can affect uptake rates (due to variations in transporter affinity) or uptake order (in rich media, *E. coli* have been observed to deplete preferred metabolites before beginning to consume others (46)). To determine whether the preferred nutrient sources in our microfluidics experiments differed with nutrient concentration, we measured the depletion of extracellular metabolites from batch cultures. Batch cultures were necessary to observe metabolite depletion, and best represented microfluidics conditions, which continuously replenished all metabolites, at the earliest time points after inoculating the media with cells.”

Thank you for highlighting our initial lack of clarity!

To clarify that we were testing for preferences in nutrient source across conditions, we also added “serine is likely the preferred metabolite across all conditions” at **Lines 245-246**.

L626: Could you provide a number in percentage?

Lines 1151-1152: we replaced “the vast majority” with “93–97% (depending on nutrient condition)” to specify the percentage of tracks belonging to single cells.

L834: The lighter lines are parallels?

The Reviewer is correct, the lighter lines in **Fig. 2a** represent length trajectories for simultaneously tracked individuals, each followed over several generations.

Line 1400-1402: we have edited the figure legend to clarify this, adding: “One line is bolded for clarity; the lighter lines are simultaneous tracks measured from different individuals.”

L839: In Figure 2a: cells not always seem to follow the nutrient signal, e.g. between 6 and 7 hours. This does not seem to mirror in 2b.

We believe the oddity that the Reviewer mentions is the irregular flattening of the exponential curve, such as observed in the bolded volume trajectory between hours 6 and 7 of **Fig. 2a**. The sudden flattening results from the cell being shifted from C_{high} to C_{low} and then gaining volume at a slower rate. In **Fig. 2a**, we visualize the volumes of selected single cells to exemplify the data used to estimate the growth rates (volumetric doubling, not cell division) visualized in **Fig. 2b**.

We believe that confusion here may have derived from the earlier questions related to cell division. We hope the revisions described earlier in this response make this distinction more clear. To cue readers into the volumetric definition of growth rate within this Figure Legend, we have added the following sentence.

Lines 1402-1403: “In the fluctuating environment, single-cell volume grows at different rates depending on the nutrient phase.”

L859: Is this an artificial term created for the interpretation of the lower growth rates of cells in fluctuating environments? If a delay in growth is the reason for the behavior of those cells, this term might not contribute to provide explanations...

We thank the Reviewer again for pointing attention to our use of Jensen’s inequality, which was already highlighted in the main text as a point of revision. Jensen’s inequality is a term used to refer to a mathematical concept (proved by Johan Jensen in Jensen *Acta Math* 1906) that states: for concave functions (such as the nonlinear relationship between nutrient concentration and growth rate), the mean of the function is always smaller than the function of the mean.

The Reviewer correctly points out that this mathematical model is not an explanation for our results and that our results deviating from Jensen’s inequality is not surprising to those familiar with growth responses (e.g. lag times, etc.). As we have described in an earlier response, we have largely revised our discussion in the Main Text to make this point more clear:

Lines 327-328: “A simple mathematical model, relevant to the concave Monod curve, illustrates a rationale that highlights the physiological implications of fluctuating nutrients.”

Lines 336-340: “This difference is consistent with the unrealistic scenario represented by Jensen’s inequality, which considers a cell that fluctuates between growth at two steady-states, G_{low} and G_{high} ($1.07 \pm 0.23 \text{ h}^{-1}$ and $2.86 \pm 0.14 \text{ h}^{-1}$, respectively). In reality, the magnitude by which G_{fluc} is lower than G_J depends on the dynamics by which single-cell growth rate responds to fluctuations in nutrient concentration.”

Overall, we feel that introducing this simple model could help readers intuit the physiological meaning of a reduction in growth (i.e., why it makes sense that temporal fluctuations at minute timescales cannot be averaged and are not averaged by cells) and hope our revisions better achieve this purpose without misleading or confusing.

Supplementary Information

SFig2: Headline does not describe a-c

Lines 48-49: We have edited the headline for **Supplementary Fig. 2** to read “Control experiments demonstrate nutrient concentration is the determinant of growth rate”. This more general headline is now true for each of the individual panels, whereas previously the headline summarized the specific controls performed in each panel.

SFig5: Last sentence: Is there no possibility to create a standard situation for these types of experiments?

We thank the Reviewer for this important question. In this comment, the Reviewer refers to a sentence that was part of the Figure Legend for the former **Supplementary Fig. 5c**, which in this revised version of the manuscript is now summarized in **Supplementary Table 8**. The sentence highlighted by the Reviewer reads:

Lines 608-610 in the Supplementary Information: “Thus, we compared growth rate across conditions performed on the same day before comparing experimental replicates.”

Given the day-to-day variability in measured steady-state growth rate, we always performed each fluctuating experiment alongside all three steady conditions (G_{low} , G_{ave} , G_{high}), which served as a standard control by which to compare the growth rate measured from fluctuating experiments (G_{fluc}) with steady-state growth rates (G_{low} , G_{ave} , G_{high}) and G_{fluc} measured from other timescales.

In setting up the experiments, we did our best to tightly replicate procedures: small OD range, similar times of day, temperature control, etc. Some of the day-to-day variability in the exact value of G_{low} , G_{ave} , and G_{high} may arise from our need to prepare nutrient solutions fresh for each experiment, potentially leading to slight concentration differences between experiments due to human error.

Another source of day-to-day variability is likely inherent to the fact that cell growth is a noisy process (**Supplementary Fig. 7**). The variation we observed between replicate measurements of G_{low} , G_{ave} , and G_{high} (**Fig. 3b**, **Supplementary Table 2**) is consistent with the noise measured from our various conditions. The noisier the growth condition, the larger the variation observed between replicate measurements in G .

Altogether, noise and variability are inherent in all measurements of single-cell growth, not only our experiments. In this study, by using three steady conditions (G_{low} , G_{ave} , G_{high}) as a standard for each fluctuating condition, we can limit the effect of this noise in our analyses (e.g. **Fig. 6b**). Further, we observe clear trends: (1) the steady-state growth rates measured from each condition are distinguishable from one another (**Supplementary Fig. 5b**) and produce a Monod curve across a wide range of nutrient concentrations, and (2) our data clearly reproduce trends established from previous work. Thus, the inherent variability in our growth rate measurements does not impede our ability to connect with already known fundamental concepts in bacterial growth and physiology.

SFig6: a: See remark main paper line 439-442.

The data are probably obtained from bulk measurements. How can this be compared to the single cell measurements shown in all other datasets?

As the Reviewer thoughtfully notes in their comment (now pertaining to **Lines 953-956** in the revised manuscript), the conditions in our microfluidics experiments are different than those in batch cultures in that nutrient in the microfluidics conditions immediately replenish, whereas nutrients can deplete in batch conditions and enable conclusions on which metabolites are preferentially consumed across our studied nutrient concentrations.

Lines 962-972 in Main Text: to better explain our experimental set-up and comparisons between batch and microfluidics conditions in the Methods, we added:

“Changes in extracellular nutrient concentrations can affect uptake rates (due to variations in transporter affinity) or uptake order (in rich media, *E. coli* have been observed to deplete preferred metabolites before beginning to consume others (41)). To determine whether the preferred nutrient sources in our microfluidics experiments differed with nutrient concentration, we measured the depletion of extracellular metabolites from batch cultures. Batch cultures were necessary to observe metabolite depletion, and best represented microfluidics conditions, which continuously replenished all metabolites, at the earliest time points after inoculating the media with cells.”

Second sentence: This assumption seems to be very artificial. Carbon source uptake is facilitated by many processes among them transport systems of different affinities, which react differently to the various concentrations in the medium. In this study, complex carbon sources were used, so I'm not convinced if this simplified assumption can be applied.

We thank the Reviewer for their careful attention to this analysis and their help in highlighting areas in which our communication about our logic and interpretations were not well explained.

To address this, we reconstructed **Supplementary Fig. 6** entirely to avoid the conceptual panels that oversimplified and misrepresented nutrient uptake in our complex conditions.

Lines 155-174 in the Supplementary Information: we now present only data from amino acids previously reported to be amongst the first metabolites consumed by *E. coli* growing in rich media (Zampieri et al. *Nature Communications*, 2019). Accordingly, the entire Figure Legend for **Supplementary Fig. 6** has been rewritten.

References cited in this response:

- Zampieri, M., Hörl, M., Hotz, F., Müller, N. F., & Sauer, U. Regulatory mechanisms underlying coordination of amino acid and glucose catabolism in *Escherichia coli*. *Nature Communications* 10, 3354 (2019).

SFig9: No growth rate values given

Unclear: the cell cycle is expected to need 12 h? Or at least 6 hours until cell division? This seems unrealistic, because cell cycling does not depend that much on nutrient conditions... It is also known for *E. coli* that replication takes about 60 min and only stays undivided if nutrient conditions are bad.

What do you mean by stabilization time? If the nutrient loss occurs within replication time, the cells obtain reserves, which they can use. This is the concept of reaching the 'critical cell size' before proliferation can start. How does this model include the 'critical cell size' concept?

We thank the Reviewer yet again for helping us catch each of these unclear elements in the previous version of our manuscript. We have revised this Figure (now **Supplementary Fig. 10**) as follows to address each of the points in this comment.

In the Figure itself, we adjusted the visualized dynamics in all panels to mirror that of **Fig. 4a** and **Fig. 6a**. We hope that this visual similarity helps readers connect that the growth rate dynamics here refer to the instantaneous growth rate dynamics measured from single shift experiments (plus some time steady at steady-state G_{low} or G_{high} , if relevant to the modeled nutrient timescale).

- We overlaid the data from Fig. 4a onto each panel to visually demonstrate how the single shift data was used to build each model.
- Note: we removed panel **a** from the original version of this Figure, as it is now part of **Fig. 6a**. The revised **Supplementary Fig. 10** now includes examples of the two models used for single-shift predictions of growth rate dynamics on faster timescales (i.e., $T = 60$ min).

Lines 351-373 in Supplementary Information: We also revised the Figure Legend to immediately remind readers that we are modeling volumetric growth over time, which is predominantly controlled by the nutrient environment and only weakly (if at all) controlled by the cell cycle (i.e., time relative to cell division) (**Supplementary Fig. 8b**).

- We avoided using "stabilization time", replacing it with more explicit phrasing: "[hours] to reach the steady-state growth rate".
- We included values on the y-axis and wrote into Legend the exact values of G_{low} and G_{high} .
- We improved our explanation on how the single shift response was used to simulate the dynamics in each model by making the Legend more specific.

Lines 418-500 in Supplementary Information: we added an additional section of text, titled "Supplementary Procedures: Construction of Null Model", which carefully details exactly how our null model (the dynamics plotted in **Supplementary Fig. 10**) were simulated.

We hope the Reviewer finds our edits to address this comment, together with our extensive edits to address earlier comments regarding our null model, a satisfyingly clear explanation of our growth rate predictions (G_{fluc}) based on single shifts.

In summary, we want to thank the Reviewer for their in-depth engagement with our manuscript, which we have really appreciated and which we feel has helped us better convey our results. By highlighting areas that were not direct or specific enough and concepts that we had not addressed in our earlier version, their review was vital in enabling us to clarify the contributions of this work and their significance.

Response to Reviewer #3

The manuscript by Nguyen and team describes an experimentally elegant analysis of growth rate variation during nutrient shifts of different temporal range (seconds to minutes). The work is novel and appears to have been well-designed and well-executed. The data produced is nicely presented and straightforward.

I do have two issues with the manuscript that would need work prior to publication in a journal such as Nat Comm.

(1) The results, such as the ones presented on Figures 3,4,5 are clear, and the interpretation of the authors is likely correct. But the authors offer no experimental evidence to demonstrate which mechanism or mechanisms is/are triggered under rapid nutrient fluctuations "novel growth physiology" derived from differential transcription, or translation, or protein turnover, or transport? Without identifying an underlying mechanism, these results only descriptive. The authors need to experimentally identify and demonstrate which mechanism is behind these growth rate fluctuations and what really causes the "novel growth physiology".

We agree with the Reviewer that the underlying molecular mechanism of this growth physiology is important in order to understand how cells achieve it; however, we believe that our study offers significant conceptual contributions even without mechanism for the following reasons:

- Single-cell studies have made path-breaking contributions to our understanding of bacterial growth, even without molecular mechanisms.
- As the first single-cell study to achieve rapid nutrient fluctuations, our study contributes a unique ecological perspective to bacterial growth. The field has long recognized that environmental fluctuations are important, yet achieving these conditions is non-trivial.

Additionally, we believe that

- A rigorous and informative study of the mechanisms of this fluctuation-induced physiology requires -omics level approaches, as cherry picking mutants in specific pathways will almost certainly affect growth.

We elaborate on each of these points in more detail below.

The lack of molecular mechanisms is not specific to our study, but rather a general limitation in the field of single-cell growth, as can be seen from several prominent papers published in the recent past (see below). **Despite the lack of molecular insight, single-cell studies have opened new doors in our understanding of bacterial growth.** For example:

- Godin et al. *Nature Methods* (2010): high-resolution measurements of cell mass determined that single cell *E. coli* grow exponentially, rather than linearly. Prior to this

method, it was unclear how single-cells added mass (populations measured by OD were much easier to measure).

- This paper offered no molecular mechanism, yet enabled all downstream work to confidently estimate single-cell growth rate as an exponential function.
- 345 citations
- Wang et al. *Current Biology* (2010): a similar microfluidic and single-cell imaging approach determined that single *E. coli* cells exhibit robustly stable growth in steady-state environments for hundreds of consecutive generations. It was previously believed that cells would age and grow slower (senescence).
 - This paper offered no molecular mechanism to describe how cells avoid senescence, yet their findings serve as critical background information to interpret molecular information when considering bacterial physiology (including the interpretation of gene expression etc.).
 - 728 citations
- Campos et al. *Cell* (2014) and Taheri-Araghi et al. *Current Biology* (2015): both of these studies use a similar microfluidic and single-cell imaging approach to determine that *E. coli* adds a constant cell size per cell cycle, providing the single-cell evidence needed for the Adder model of cell size homeostasis.
 - Neither paper determines the molecular underpinnings of the Adder model, yet these works have launched a massive body of work that has since found evidence of the Adder model across diverse bacteria and conditions. The Adder model is now the most accepted model of cell size control (Willis and Huang *Nature Reviews Microbiology* 2017)
 - 338 and 490 citations, respectively

These works provided important conceptual foundations for later studies to pursue for mechanistic insights. For example, three very recent single-cell growth studies have proposed different mechanisms for the Adder model (Wallden et al. *Cell* 2016; Harris and Theriot *Cell* 2016; Si et al. *Current Biology* 2019). Each of these studies hone in on a specific hypothesis to describe the mechanism of cell size homeostasis, and several additional studies have sought to reconcile the different mechanisms posed by each model (Amir *eLife* 2017; Micali et al. *Cell Reports* 2018a; Micali et al. *Science Advances* 2018b). To date, there are still no broadly accepted molecular mechanisms of how bacterial cells coordinate their cell size and growth rate, yet there is no question how important these single-cell works have been in advancing the field (Willis & Huang *Nature Reviews Microbiology* 2017).

Our study is a path-breaking advance in that we directly observe single-cell growth in fluctuations for the first time, providing the first single-cell insights into growth under conditions that capture a fundamental aspect of the natural environment of many microorganisms, namely the temporal variability in nutrient availability. Each of the previously mentioned examples was path-breaking because of the new insights unique to single-cell observations; we believe that our study falls on the same line by introducing a totally unexplored and ecologically relevant environmental factor -- fluctuations. Our conceptual findings and our unique dataset offer the first connection between the above works (cell cycle

control under steady-state growth/ homeostatic conditions) and bacterial growth in natural environments, in which single cells routinely experience fluctuations.

Imposing carefully controlled environmental fluctuations while observing single-cell properties is far from trivial. The methods reported in our manuscript represent the solutions we found after nearly 3 years of attempting to gather uncompromised single-cell growth data in fluctuations. Our novel microfluidic system was designed to automate precise fluctuations while ensuring that the cells we analyzed experienced the exact signal we wanted to deliver, unaffected by diffusional smearing, nutrient consumption and metabolite secretions from other cells. We opted for our final set-up (surface-attached cells) after considering various alternatives, in which the cells were either not confined enough in space (such that cell movement impeded long-term tracking) or too confined (potentially compromising size measurements or our knowledge of the nutrient environment they experienced). Surface-attached growth also required troubleshooting, as surface-attachment is generally associated with biofilm or biofilm-like growth. Trial and error enabled us to determine unexpected protocol requirements to limit the growth of bacterial aggregates and obtain clean single-cell data in our microchannel experiments. We are pleased that the Reviewer considers our work as an “experimentally elegant analysis” and would like to emphasize that achieving this was a considerable feat.

Finally, because growth physiologies are inherently complex and produced by several highly integrated cellular processes, **we believe a rigorous and informative study of the mechanisms of this fluctuation-induced physiology requires -omics level approaches.** Certain genes are clear candidates for involvement in *E. coli*'s response to nutrient fluctuations, such as *relA* and *spoT*, which synthesize the starvation signal (p)ppGpp. However, because these candidates regulate growth rate (Potrykus et al. *Environmental Microbiology* 2011), mutants have different growth rates than wild-type cells even in steady environments. Thus, how specific genes and factors may regulate a fluctuation-adapted physiology would be confounded by differences that already exist between mutants and wild-type in steady and single-shift conditions. Microfluidic experiments with mutant strains will undoubtedly be useful; however, we believe these experiments should be paired with -omics information to tease out how specific genes or pathways interact differently between single shifts vs. fluctuating environments.

Reviewer 1 has acknowledged this point explicitly, writing in regard to uncovering mechanisms: “that work likely would require an entirely separate study.”

Works cited in this response:

- Amir, A. Point of view: Is cell size a spandrel?. *eLife*, 6, e22186 (2017).
- Campos, M., Surovtsev, I. V., Kato, S., Paintdakhi, A., Beltran, B., Ebmeier, S. E., & Jacobs-Wagner, C. A constant size extension drives bacterial cell size homeostasis. *Cell* 159(6), 1433-1446 (2014).
- Harris, L. K., & Theriot, J. A. Relative rates of surface and volume synthesis set bacterial cell size. *Cell* 165(6), 1479-1492 (2016).

- Micali, G., Grilli, J., Marchi, J., Osella, M., & Lagomarsino, M. C. Dissecting the control mechanisms for DNA replication and cell division in *E. coli*. *Cell Reports* 25(3), 761-771 (2018).
- Micali, G., Grilli, J., Osella, M., & Lagomarsino, M. C. Concurrent processes set *E. coli* cell division. *Science Advances*, 4(11), eaau3324 (2018).
- Potrykus, K., Murphy, H., Philippe, N. & Cashel, M. ppGpp is the major source of growth rate control in *E. coli*. *Environmental Microbiology*, 13(3), 563-575 (2011).
- Taheri-Araghi, S., Bradde, S., Sauls, J. T., Hill, N. S., Levin, P. A., Paulsson, J., ... & Jun, S. Cell-size control and homeostasis in bacteria. *Current Biology* 25(3), 385-391 (2015).
- Wallden, M., Fange, D., Lundius, E. G., Baltekin, Ö., & Elf, J. The synchronization of replication and division cycles in individual *E. coli* cells. *Cell* 166(3), 729-739 (2016).
- Wang, P., Robert, L., Pelletier, J., Dang, W. L., Taddei, F., Wright, A., & Jun, S. Robust growth of *Escherichia coli*. *Current Biology* 20(12), 1099-1103 (2010).
- Willis, L., & Huang, K. C. Sizing up the bacterial cell cycle. *Nature Reviews Microbiology*, 15(10), 606-620 (2017).

(2)The other issue I have with this manuscript has to do with the expectation that growth physiology derived from single nutrient based steady-state models has to match with growth physiology under rapid nutrient shifts. The authors state "These results demonstrate that growth rate in rapid fluctuations is qualitatively and quantitatively distinct from steady-state growth dynamics." This result/conclusion is really not surprising as either their cells will be experiencing a completely different steady-state (if that has been reached) or still be under transient conditions, that is, adapting to the shifts. I think the authors could instead of presenting this fact as something completely unexpected and profound, present it as an obvious condition derived from shifting bacteria from one condition to another.

We respectfully push back against the notion that our manuscript presents no conceptual advance and that the findings we present were "obvious", and we have sought to further clarify this in our revision.

We argue that our study is indeed of broad conceptual novelty and interest. As the Reviewer acknowledges in the general remarks, our study is the first to consider the effect of nutrient fluctuations on these short timescales, the first to report this novel growth physiology, and the first to systematically study fluctuations at the single-cell level. Reviewers 1 and 2 found the work "exciting", "impressive" and "appealing" and that our main finding "opens unusual views" and "stimulates thinking".

Reviewer 1 wrote: "This experiment presents an exciting hypothesis about the adaptations of microorganisms to real environments where nutrient availability likely fluctuates due to numerous sources of environmental variation. The large sustained growth rate detriment that was observed under fluctuating environments would present a selection pressure that would drive adaptive mechanisms to maintain homeostasis despite fluctuations. Thus, the whole

scenario seems quite plausible and appealing” and that “on the whole, the work is very impressive and the results are quite clear.”

Reviewer 2 wrote: “Nguyen and coworkers addressed the very interesting topic of how a bacterial cell grows under fluctuating nutrient conditions” adding that “the approach is a very mechanistic one but opens unusual views on cell growth and stimulates thinking on bacterial cell growth, which is still a much discussed topic.”

In response to the Reviewer’s comment that our “result/conclusion is not really surprising”, we disagree that our results were totally expected. We argue that had different growth rate responses under fluctuations been expected by the field, single shift experiments would not have remained the experimental model by which the field studies growth in fluctuations, including very recently (Erickson et al. *Nature* 2017; Mori et al. *Nature Communications* 2017; Kohanim et al. *Cell Reports* 2018; Basan et al. *Nature* 2020). Even state-of-the-art work by top researchers, published recently in *Nature*, claims to study fluctuations but in fact studies single shifts (Basan et al. *Nature* 2020).

Given the broad acceptance of single shifts as a model for fluctuations, we argue that the expectation would have been for cells in fluctuating environments to respond to each repeated change in nutrient just as if it were a single shift from steady state. By this expectation, the response to fluctuations would have been directly predictable from the measured response in single upshifts and downshifts. **Our work shows that this obvious expectation is in fact not realized**. Thus, we argue that our observation of a response to fluctuations that differs from the response to single shifts is novel and unexpected. We appreciate a comment from Reviewer 1, who wrote that evolutionary pressures suggest that our interpretations of this response are “quite plausible and appealing”. Our results indeed conceptually fit into a greater ecological framework, the idea that bacteria are adapted to grow amidst prominent features of their environment. We believe this consistency does not deem our finding “obvious” but rather emphasizes the likelihood that adaptations to rapid fluctuations are widespread across bacteria, well beyond our model organism *E. coli*.

We thank the Reviewer for their feedback, which enabled us to clarify the contributions of this work and their significance. For example, we have edited the statement highlighted by the Reviewer by replacing “steady-state growth dynamics” with “single-shift dynamics” to be explicit when we are comparing fluctuations not to steady environments but to single shifts between steady states. This particular example is on **Line 765** of the revised manuscript.

References cited in this response:

- Erickson, D. W., Schink, S. J., Patsalo, V., Williamson, J. R., Gerland, U., & Hwa, T. A global resource allocation strategy governs growth transition kinetics of *Escherichia coli*. *Nature* 551(7678), 119 (2017).
- Mori, M., Schink, S., Erickson, D. W., Gerland, U., & Hwa, T. Quantifying the benefit of a proteome reserve in fluctuating environments. *Nature Communications*, 8(1), 1225 (2017).

- Kohanim, Y. K., Levi, D., Jona, G., Towbin, B. D., Bren, A., & Alon, U. A bacterial growth law out of steady state. *Cell Reports* 23(10), 2891-2900 (2018).
- Basan, M., Honda, T., Christodoulou, D., Hörl, M., Chang, Y.F., Leoncini, E., ... Sauer, U. A universal trade-off between growth and lag in fluctuating environments. *Nature* 584(7821), 470-4 (2020).

REVIEWERS' COMMENTS

Reviewer #1 (Remarks to the Author):

I only had minor comments and suggestions for the initial submission, and these have been addressed by the authors in their revisions and rebuttal. I have no further concerns. Congratulations to the authors on this nice work

Reviewer #2 (Remarks to the Author):

The authors have answered all questions in detail and to the point. The appropriate changes have been made in the main text and in the supplementary information. I have no further questions or comments. I recommend the acceptance of the study.

Reviewer #3 (Remarks to the Author):

The authors did a good job reviewing the manuscript and further clarifying several issues that were not clear in the first draft. The rebuttal letter was also very good.

RESPONSE TO REVIEWER COMMENTS (after revision)

The Reviewers' comments are copied below in black. Our responses are in blue.

Response to Reviewer #1

I only had minor comments and suggestions for the initial submission, and these have been addressed by the authors in their revisions and rebuttal. I have no further concerns. Congratulations to the authors on this nice work

We thank the Reviewer again for their positive assessment of our work and for the comments and suggestions that helped us enrich the work in our revisions.

Response to Reviewer #2

The authors have answered all questions in detail and to the point. The appropriate changes have been made in the main text and in the supplementary information. I have no further questions or comments. I recommend the acceptance of the study.

We are pleased to hear that the Reviewer found our revisions detailed and to the point. We are grateful for the Reviewer's detailed questions, which helped deepen the information and discussion presented in our study.

Response to Reviewer #3

The authors did a good job reviewing the manuscript and further clarifying several issues that were not clear in the first draft. The rebuttal letter was also very good.

We are pleased that the Reviewer found the clarity of our revised manuscript much improved from our initial submission and thank the Reviewer for their comments, which enabled us to identify areas to improve clarity. We thank the Reviewer for their positive assessment of our rebuttal and grateful for the opportunity it gave us to lay out the thoughts within it in writing.